# SLiM-LLM: Salience-Driven Mixed-Precision Quantization for Large Language Models

## Abstract

Large language models (LLMs) have achieved remarkable progress, but their extensive number of parameters results in high memory usage, significant loading latency, and substantial computational demands. To address these challenges, post-training quantization (PTQ) has emerged as an effective technique for compressing model weights. In the context of PTQ for LLMs, existing uniform quantization methods, though efficient in terms of memory and computational requirements, often struggle to maintain performance. In this paper, we propose **SliM-LLM**, a Salience-Driven Mixed-Precision Quantization scheme that achieves group-wise bit-width allocation with mixed precisions for efficient LLMs with high accuracy. Building on our observation that salient/important weights often follow a structured distribution, we incorporate two core components to preserve post-quantization performance in LLMs while maintaining efficiency: **1)** *Salience-Determined Bit Allocation* adaptively assigns bit widths to groups within each layer based on their group-level salience, aiming to minimize the reconstruction error of activations; and **2)** *Salience-Weighted Quantizer Calibration* optimizes quantizer parameters by incorporating element-level salience, ensuring that the most critical weights are preserved, further preserving important weights information. With its structured group partitioning, SliM-LLM offers a hardware-friendly quantization approach, maintaining computational and memory efficiency comparable to highly optimized uniform quantization methods. Extensive experiments demonstrate that SliM-LLM significantly improves the accuracy of various LLMs when quantized to ultra-low bit widths. For instance, a 2-bit quantized LLaMA-7B model achieves nearly 6x memory reduction compared to its floating-point counterpart, alongside a 48% reduction in perplexity compared to the leading gradient-free PTQ method, all while maintaining GPU inference speed. Furthermore, SliM-LLM+, which incorporates gradient-based quantizers, reduces perplexity by an additional 35.1%.

## 1 Introduction

Large language models (LLMs) have exhibited exceptional performance across a wide array of natural language benchmarks (Brown et al., 2020; Hendrycks et al., 2020). Notably, LLaMA (Touvron et al., 2023a) and GPT (Brown et al., 2020) series have significantly contributed to the ongoing evolution of LLMs towards universal language intelligence. The powerful language understanding capabilities of LLMs have been transferred to multi-modal domains (Li et al., 2024b; Achiam et al., 2023; Team et al., 2023; Zhang et al., 2023; Huang et al., 2024b), laying the foundation for artificial general intelligence (AGI) (Bubeck et al., 2023). Despite these significant achievements, the substantial computational and memory requirements of LLMs pose huge challenges for real-world applications, particularly in resource-constrained environments.

To address resource constraints of LLMs, post-training quantization (PTQ) has emerged as an efficient yet effective compression technique (Dettmers et al., 2022), showing success in quantizing the weights of pre-trained LLMs (Frantar et al., 2022; Lin et al., 2023; Shao et al., 2023; Lee et al., 2023; Chee et al., 2024). As LLMs continue to scale, the demand for more aggressive low-bit compression becomes critical due to limited computational and storage resources in application (Huang et al., 2024a; Tseng et al., 2024). However, significant performance degradation remains a challenge in low-bit scenarios ($\leqslant$ 3-bit). To mitigate this, unstructured mixed-precision quantization (Shang et al.,

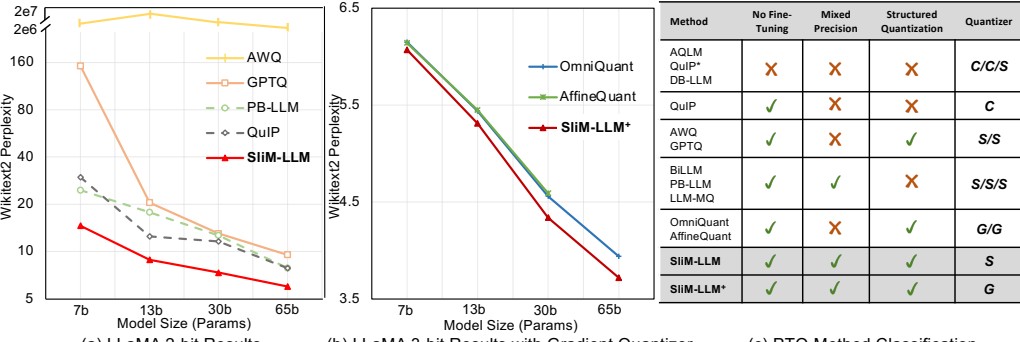

Figure 1: (a) The perplexity (↓) of existing low-bit PTQ methods of LLaMA at 2-bit. Solid-line indicates methods with structured quantization group. (b) Compare PTQ methods with gradient quantizer at 3-bit. (c) Features of current low-bit quantization methods. **C** denotes codebook-based, **S** is statistic-based, and **G** represents gradient-based quantizers.

2023; Huang et al., 2024a; Dettmers et al., 2023) and vector quantization (Chee et al., 2024; Tseng et al., 2024; Egiazarian et al., 2024) methods have been developed to preserve performance. While these approaches have advanced the field, they are often hardware-unfriendly, introducing extra storage requirements such as storing bitmaps or code indices, along with additional computations for vector decoding. This creates a bottleneck, limiting further reductions in memory and computational demands during deployments. In sum, ensuring the accuracy of LLMs while maintaining efficiency during deployment remains a significant challenge for current PTQ approaches.

This paper presents a **Sali**ence-Driven **M**ixed-Group LLM (**SliM-LLM**) framework, an accurate and inference-efficient PTQ method for LLMs (⩽ 3-bit). Our approach is grounded in the key observation that *salient or important weights, which are critical to model performance, exhibit a structured distribution, often clustering within certain channels* (see Sec. 3.2.1 and Fig. 3). This insight, largely overlooked by prior research (Frantar et al., 2022), forms the basis for designing SliM-LLM as a structured, hardware-friendly mixed-precision low-bit quantization method. It preserves performance through two key designs that retain important weights at both the global group and local element levels. First, we develop a novel *Salience-Determined Bit Allocation* (SBA) method, which adaptively assigns bit-widths to each quantization group based on their group-level salience ranking. The allocation strategy is optimized to reduce activation reconstruction errors. By applying higher precision to more important groups and reducing the bit-width for less critical ones, SBA achieves a low average bit-width while enhancing the overall performance of LLMs. Next, we introduce the *Salience-Weighted Quantizer Calibration* (SQC), which enhances sensitivity to locally salient weights, ensuring that critical information within groups is preserved. SQC works collaboratively with SBA, exploiting the local and global salience of weights to preserve the performance of LLMs after quantization. Unlike element-wise mixed-precision methods (Shang et al., 2023; Dettmers et al., 2023; Huang et al., 2024a), SliM-LLM is inherently structured, eliminating additional bit or computational overhead while preserving high performance. This is further demonstrated through our deployment of SliM-LLM in an application-level inference tool [1] for LLMs, enabling efficient mixed-precision inference on GPUs with consistently strong performance.

Experiments show that for various LLM families, SliM-LLM surpasses existing training-free PTQ methods on diverse benchmarks, particularly in low-bit scenarios. Using GPTQ as the backbone, SliM-LLM improves the perplexity scores of 2-bit LLaMA-13B and LLaMA2-13B on WikiText2 (Merity et al., 2016) from 20.44 and 28.14 to 8.87 and 9.41, denoting performance improvements of over 56%, respectively. SliM-LLM even outperforms other element-wise mixed-precision PTQ methods, such as PB-LLM (Shang et al., 2023), APTQ (Guan et al., 2024) and LLM-MQ (Li et al., 2024a), in a deployment-friendly manner, showcasing its superior low-bit accuracy and efficiency. We also integrate SliM-LLM into OmniQuant (Shao et al., 2023) and obtain SliM-LLM$^+$ through gradient optimization to further improve quantization quality. Moreover, the group-wise mixed-precision strategy can smoothly be adapted to existing quantization-aware training (QAT) (Liu et al., 2023), fine-tuning based (Guo et al., 2023; Liao & Monz, 2024; Dettmers et al., 2024), or codebook-based (Chee et al., 2024; Egiazarian et al., 2024; Tseng et al., 2024) LLMs compression methodologies. The structure of weight salience we theoretically identify introduces a new practical view of the weight quantization of LLMs.

---

[1] https://github.com/AutoGPTQ/AutoGPTQ

Figure 2: Illustration of our proposed SliM-LLM. The *Salience-Determined Bit Allocation* (SBA) optimizes activation-aware structured precision, optimizing the global information distribution in quantization. *Salience-Weighted Quantizer Calibration* (SQC) detects discretely distributed salient weights, enhancing the local important information in LLMs.

## 2 RELATED WORK

**Large Language Models** (LLMs) have been significantly developed in diverse natural language processing domains, establishing a prominent paradigm in these fields (Bubeck et al., 2023; Chang et al., 2024; Zhao et al., 2023; Brown et al., 2020; Touvron et al., 2023a). Nevertheless, the exceptional success of LLMs depends on massive parameters and computations, posing significant challenges for deployment in resource-constrained environments. Consequently, research into the compression of LLMs has emerged as a promising field. Existing compression techniques for LLMs primarily include low-bit quantization, pruning, distillation, and low-rank decomposition (Xu et al., 2023; Ganesh et al., 2021; Frantar et al., 2022; Xiao et al., 2023a; Shao et al., 2023; Chee et al., 2024; Zhu et al., 2023; Frantar & Alistarh, 2023; Huang et al., 2024a; Qin et al., 2024). Among these technologies, low-bit quantization gains remarkable attention, for efficiently reducing the model size without change of network structure(Zhu et al., 2023; Zhao et al., 2023; Chang et al., 2024).

**Quantization of LLMs** can be generally divided into QAT and PTQ. QAT, by employing a retraining strategy based on quantized perception, better preserves the performance of quantized models. LLM-QAT (Liu et al., 2023; Ma et al., 2024a) addresses the data obstacle issue in QAT through data-free distillation. However, for LLMs with huge size of parameters, the cost of retraining is extremely inefficient(Chang et al., 2024). Therefore, PTQ has become a more efficient choice for LLMs. For instance, LLM.int8() (Liu et al., 2023) and ZeroQuant (Yao et al., 2022) explore the quantization strategies for LLMs in block-wise, which is a low-cost grouping approach that reduces hardware burden. Subsequently, AWQ (Lin et al., 2023) and OWQ (Lee et al., 2023) also propose scaling transformations on outlier channels of weight to preserve their information representation capacity. GPTQ (Frantar et al., 2022) reduces the group quantization error of LLMs through Hessian-based error compensation (Frantar & Alistarh, 2022), achieving commendable quantization performance at 3-bit. OmniQuant (Shao et al., 2023) introduces a learnable scaling quantizer to reduce quantization errors in an output-aware manner. To achieve LLM quantization at ultra-low bit-width, recent novel efforts such as QuIP (Chee et al., 2024), QuIP# (Tseng et al., 2024), and AQLM (Egiazarian et al., 2024) promote quantization performance at 2-bit through matrix decomposition with learnable codebooks and fine-tuning. Recent studies (Qin et al., 2024; Liao & Monz, 2024; Dettmers et al., 2024; Guo et al., 2023) have further refined compression techniques by integrating post-training quantization (PTQ) with parameter-efficient fine-tuning (PEFT) to enhance model performance via additional parameter learning.

**Mixed-Precision Quantization** exploits variations in the importance and redundancy of model parameters, assigning different bit-widths to each component. HAWQ V2 (Dong et al., 2020) and V3 (Yao et al., 2021) optimize bit-width allocation layer-wise in traditional visual networks through Hessian analysis and Integer Linear Programming (ILP). Alternatively, OMPQ (Ma et al., 2023) employs network orthogonality instead of Hessian for similar purposes. In LLMs, APTQ (Guan et al., 2024) extends HAWQ's strategy, allocating varied bit-widths to different transformer blocks based on Hessian-trace, thus improving the accuracy of 3-bit LLMs. However, such block-wise or layer-wise mixed-precision allocation at 2-bit still fails to maintain post-compression performance in LLMs. Recent studies such as SpQR (Dettmers et al., 2023), PB-LLM (Shang et al., 2023), and LLM-MQ (Li et al., 2023) have introduced finer-grained partitioning for grouped quantization with element-wise mixed-precision for accurate weight quantization. Nevertheless, these low-bit methods still rely on special structures and fine-grained grouping to ensure accuracy, which brings the huge burden of real hardware deployment and inference speed.

## 3 SLIM-LLM

This section introduces a group-wise mixed-precision quantization method, SliM-LLM, designed to overcome the accuracy and efficiency bottlenecks of mixed-precision LLMs. We devise two novel strategies for LLMs, including the use of *Salience-Determined Bit Allocation* (SBA) based on global salience distribution to determine group bit-widths, and *Salience-Weighted Quantizer Calibration* (SQC) to enhance the perception of locally important weight information. We introduce SBA and SQC in Sec. 3.2 and Sec. 3.3, respectively.

### 3.1 PRELIMINARIES

**Quantization Framework.** We first present the general uniform quantization process of LLMs according to common practice (Liu et al., 2023; Shao et al., 2023; Achiam et al., 2023). The quantization process requires mapping float-point weights distributed within the interval $[w_{\min}, w_{\max}]$ to an integer range of $2^N$, where $N$ is the target bit-width. The quantization function for weight $\boldsymbol{w}_f \in \mathbb{R}^{n \times m}$ follows:

$$\hat{\boldsymbol{w}}_q = \text{clamp}(\lfloor \frac{\boldsymbol{w}_f}{s} \rceil + z, 0, 2^N - 1), \; s = \frac{w_{\max} - w_{\min}}{2^N - 1}, \; z = -\lfloor \frac{w_{\min}}{s} \rceil \quad (1)$$

where $\hat{\boldsymbol{w}}_q$ indicates quantized weight which is integer, $\lfloor \cdot \rceil$ is round operation and $\text{clamp}(\cdot)$ constrains the value within integer range (e.g. $[0, 1, 2, 3]$, $N = 2$). $\Delta$ is scale factor and $z$ is quantization zero point, respectively. When converted to 1-bit quantization, the calculation follows:

$$\hat{\boldsymbol{w}}_b = \text{sign}(\boldsymbol{w}_f), \; \text{sign}(w) = \begin{cases} 1 & \text{if } w \geq 0, \\ -1 & \text{others.} \end{cases}, \; \alpha = \frac{1}{l} ||\boldsymbol{w}_f||_{\ell 1} \quad (2)$$

where $\hat{\boldsymbol{w}}_b$ is binary result. $\alpha$ denotes binarization scales and $l$ is the number of elements in weight (Qin et al., 2023), used for dequantization through $\alpha \hat{\boldsymbol{w}}_b$. We can formalize the per-layer loss in PTQ, following the common practice (Nagel et al., 2020; Frantar et al., 2022):

$$\mathcal{L}(\hat{\boldsymbol{w}}_f) = ||\boldsymbol{x}\boldsymbol{w}_f^\top - \boldsymbol{x}\hat{\boldsymbol{w}}_f^\top||^2 \approx \text{tr}((\hat{\boldsymbol{w}}_f - \boldsymbol{w})\boldsymbol{H}(\hat{\boldsymbol{w}}_f - \boldsymbol{w})^\top) \quad (3)$$

where $\boldsymbol{x} \in \mathbb{R}^{t \times m}$ denotes the input vectors from calibration dataset, $\hat{\boldsymbol{w}}_f \in \mathbb{R}^{n \times m}$ is dequantized weight from quantization result in Eq. (1) or Eq. (2), and $\boldsymbol{H} = \frac{1}{P} \sum_{k=1}^{P} \boldsymbol{x}^{[k]^\top} \boldsymbol{x}^{[k]}$ is proxy Hessian matrix by Levenberg-Marquardt approximation (Marquardt, 1963; Frantar & Alistarh, 2022) from a set of input activations.

**Parameter Salience.** In LLMs, the importance of each element in the weight matrix is various (Dettmers et al., 2023; Frantar & Alistarh, 2023). According to Eq. (3), quantizing different elements causes different impacts on the model's output loss. Elements that significantly influence the loss are termed salient weights. Consequently, we follow the SparseGPT (Frantar & Alistarh, 2023) to define the salience of each element as:

**Definition 1.** *In the quadratic approximation of the loss as expressed in Eq. (3), we give the Hessian matrix $H \in \mathbb{R}^{m \times m}$ generated by $\frac{1}{P} \sum_{k=1}^{P} \boldsymbol{x}^{[k]^\top} \boldsymbol{x}^{[k]}$ for a weight matrix, the removal of the element at $(i, j)$ induces an error $\delta_{i,j} = \frac{w_{i,j}^2}{[\boldsymbol{H}^{-1}]_{j,j}^2}$ to the output matrix for linear projection in LLMs.*

where $[\boldsymbol{H}^{-1}]_{jj}$ denotes the $j^{th}$ diagonal entry for the inverse Hessian, and $\boldsymbol{H}^{-1}$ can be efficiently calculated through Cholesky decomposition (Krishnamoorthy & Menon, 2013). According to Definition. 1, we map the elimination error $\delta_{ij}$ to the salience measure of each weight element in LLMs, representing the impact of different weights on the output loss and the language capabilities, which also leads the generation of mixed-precision quantization strategies (Dettmers et al., 2023; Shang et al., 2023; Huang et al., 2024a; Li et al., 2024a) for LLMs. However, existing mixed-precision solutions require the discrete allocation of bit-widths across the entire weight matrix, which imposes a significant burden on hardware computations, thereby affecting the inference efficiency.

### 3.2 SALIENCE-DETERMINED BIT ALLOCATION

We reveal the phenomenon of spatial clustering in the distribution of weight salience, which inspires our proposed concept of group-wise mixed-precision quantization for LLMs, and then introduce the *Salience-Determined Bit Allocation* (SBA) technique to allocate the optimal precision to each group.

### 3.2.1 SPATIAL DISTRIBUTION OF GLOBAL SALIENCE

We first conduct an empirical investigation into the weight salience distribution. The results reveal that certain channels exhibit higher salience and show tendencies for spatial clustering. As illustrated in Fig. 3, salient clustering are identified around the $2100^{th}$, $3218^{rd}$ and $3853^{rd}$ channels within the $2^{nd}$ layer's attention projection of the LLaMA-7B model. A similar structured pattern is observed near the $600^{th}$, $2200^{th}$ and $3992^{nd}$ channels in the $10^{th}$ layer. Also, clustered salience is detected in other layers (as shown in Fig. 3). More examples of spatial clustering of salience are provided in Appendix G.

Then, we analyze the underlying reasons for this phenomenon. According to Definition 1, the salience of weights is proportional to the magnitude of the weights and the trace of the Hessian matrix, which can be approximated by the product of input activations $x^\top x$. In

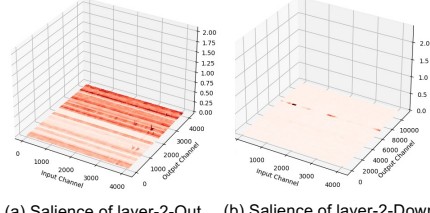

(a) Salience of layer-2-Out    (b) Salience of layer-2-Down

(c) Salience of layer-10-Out    (d) Salience of layer-10-Down

Figure 3: Salience weight distribution in layer-2 and layer-10 of LLaMA-7B.

LLMs, activations exhibit extreme outlier channels, while the numerical differences in weights are relatively slight (Xiao et al., 2023a; Nrusimha et al., 2024). Therefore, we propose an analysis of how the outlier channels in activations influence the distribution of weight salience:

**Theorem 1.** *Given the input calibration activation $x \in \mathbb{R}^{t \times m}$ with an outlier channel $x^*_{:,p} \gg x_{:,j}, \forall j \in [0, m], j \neq p$ at the position of channel-p. The trace elements of $H = x^\top x$ will show great outlier value at $(p, p)$, where $H_{p,p} \gg H_{j,j}, \forall j \in [0, m], j \neq p$, as $H_{p,p}$ is produced by $[x^{*\top}_{:,p} x^*_{:,p}] = \sum_{i=0}^{t} x^{*2}_{i,p}$, which further leads to the parameter salience larger at the $p^{th}$ channel of weight, where $\delta_{:,p} > \delta_{:,k}, \delta_{:,k} = \frac{w^2_{:,k}}{[H^{-1}]^2_{k,k}}, \forall k \in [0, t], k \neq p$.*

Theorem 1 elucidates the influence of outlier activation on the distribution of channel salient weights (detailed proof in the Appendix G.1). Furthermore, recent research indicates that outlier channels in LLMs activations consistently appear in fixed yet clustered patterns (Nrusimha et al., 2024). According to Theorem 1, these consistently occurring anomalous activations result in the distribution of salient weights, as depicted in Fig. 3. Then, during group-wise quantization, the average salience of each group shows different features.

Meanwhile, previous unstructured mixed-precision, incurred additional storage requirements and computational overheads, affecting the real-time inference. However, the strong spatial structured characteristics observed in the salient of weights in this section strongly inspire us to first develop a group-wise mixed-precision strategy within the weight matrix while maintaining inference efficiency. Therefore, we aim to allocate bit-widths based on intra-group salient disparities, which not only enhances quantization accuracy but also ensures the inference efficiency of LLMs with structured bit-widths saving and dequantization.

### 3.2.2 SALIENCE-DETERMINED BIT ALLOCATION FOR STRUCTURED GROUP

To allocate optimal bit-widths to each group, we introduce a *Salience-Determined Bit Allocation* (SBA) technique for mixed-precision LLMs, as depicted in Fig. 2. This technique, predicated on the differences in group salience, determines the optimal bit-width allocation for different groups by minimizing the distance of information entropy with the original weight output.

Specifically, we first utilize the average salience as the importance indicator for each weight group and rank them accordingly. The proposed SBA optimizes the following formula to determine the optimal number of salient-unsalient quantization groups of LLMs:

$$\text{Objective}: \arg\min \mathcal{D}_{kl}\left(xw_f^\top \,\|\, x(\hat{w}_{sba})^\top\right), \, \hat{w}_{sba} = [\hat{w}_{0,b_0}, \hat{w}_{1,b_1}...\hat{w}_{k-1,b_{k-1}}, \hat{w}_{k,b_k}]$$
$$\text{Constrain}: |\mathcal{G}_{N-1}| = |\mathcal{G}_{N+1}|, \, \mathcal{G}_{N-1} = \{b_i | b_i = N - 1\}, \, \mathcal{G}_{N+1} = \{b_j | b_j = N + 1\},$$
(4)

where $\mathcal{D}_{kl}(\cdot\|\cdot)$ denotes the Kullback-Leibler (KL) divergence between two outputs, $\hat{w}_f^{sba}$ generally represents the de-quantization results of weight, employing group-wise mixed-precision designated

as $[\hat{\boldsymbol{w}}_{0,b_0}, \hat{\boldsymbol{w}}_{1,b_1}...\hat{\boldsymbol{w}}_{k-1,b_{k-1}}, \hat{\boldsymbol{w}}_{k,b_k}]$, where $b_i$ represents the bit-width for the $i^{th}$ group and $\mathcal{G}$ is a set of groups with the same bit-width, $N$ is the targeted average bit-width. We apply a compensation constraints strategy to maintain a consistent average bit-width for our SBA. For example, in 2-bit quantization, the groups with the highest salience are quantized to 3-bit. To offset the additional bits, we quantize an equal number of groups with the lowest salience to 1-bit ($|\mathcal{G}_{N-1}| = |\mathcal{G}_{N+1}|$), while the remaining groups are set to 2-bit.

We utilize an effective double-pointer search (more detailed examples in Appendix C) to optimize our objective in Eq. (4). When the weight output channel size is $m$ and group size is 128, $k = \frac{m}{128}$, the search region for weight is limited to $[0, \frac{k}{2}]$, which is highly efficient with limited searching space, *e.g.*, only 16 iterations are needed in LLaMA-7B. We also provide detailed searching error examples in Appendix C. Notably, SBA diverges from traditional quantization with mean squared error (MSE) in Eq. (3) by instead utilizing the KL divergence as its measure of loss. Compared to using the mean squared error (MSE) for weights, SBA leverages the KL divergence of block outputs as a precision allocating metric, aiming to maximize the similarity between the distribution of the LLM's output activation matrix and the quantized activation distribution. This approach enhances the model's information representation capacity under low-bit quantization, facilitating optimal bit-width allocation. We note that HAWQ v2 (Dong et al., 2019) employs ILP to allocate bit-width for layers, which can also be adapted to our group-wise target. However, unlike the allocation of precision based solely on the weight matrix loss of each group, SBA can accurately perceive the impact of different precisions within each block on the model's output information, allowing for a more optimal bit-width allocation. More experiments comparing SBA and ILP are shown in Section 4.2.

### 3.3 Salience-Weighted Quantizer Calibration

In addition to the global group-wise distribution of salience, we notice that salience within the group still shows local differences in discrete distribution. Common existing quantizers apply uniform consideration across all weights to minimize the effect (error) of quantization, lacking the capability to perceive differences in local salience. Therefore, in this section, we introduce a *Salience-Weighted Quantizer Calibration* (SQC) to enhance the information of significant weights within the group by amplifying the quantizer awareness of salient weight.

#### 3.3.1 Discrete Distribution of Local Salience

In the aforementioned section, we group-wisely allo-cate the bit-width for each group based on the global salience. To maintain the efficiency of quantized in-ference, we employ a commonly used sequential struc-tured grouping (Frantar et al., 2022; Lin et al., 2023; Shao et al., 2023). However, this group-wise mixed-precision also leads to differences in salience among the various elements within the same group. Specifically, as the salience distribution in Fig. 4, within the $10^{th}$ attention output layer of LLaMA-7b, a subset of sparse weights within the comparatively less salient Group-2 (Fig. 4) still maintains a high level of importance. In LLMs, a small number of weight elements with outliers affect the local distribution of salience. These discrete weights typically account for only approximately 1% within the group but play a crucial role in the modeling capability of LLMs.

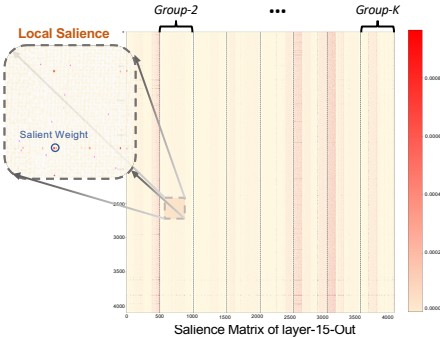

Figure 4: Local salience distribution of the $10^{th}$ MHA output layer in LLaMA-7B.

The existing vanilla quantizers face the challenge of representing significant weight information, by only considering the mean error of all elements within a group. When quantizing weights according to Eq. (1) in group-wise format, a large number of non-salient weights at the intra-group statistical level tend to dominate the parameters generated by the quantizer. This leads to a degradation of salient information within the group, thereby affecting the model performance of LLMs.

### 3.3.2 SALIENCE-WEIGHTED QUANTIZER CALIBRATION FOR LOCAL SALIENCE AWARENESS

To prevent the degradation of local salient weight information in each group, we propose the *Salience-Weighted Quantizer Calibration* (SQC), which enhances the expression of salient weights through local salience awareness, thereby reducing the quantization error of these significant elements and improving the compressed performance of LLMs.

Based on a common observation (Dettmers et al., 2023; Huang et al., 2024a), the proportion of relatively salient weights in each group is only 1-5%. Therefore, we employ the 3-$\sigma$ rule for a mask to select the salience part ($\boldsymbol{w} < (\mu - 3\sigma) \cup \boldsymbol{w} > (\mu + 3\sigma)$) in each group (Fig. 2), which accounts for about 1% elements. After the selection, we get $\boldsymbol{w}_i = \boldsymbol{w}_i^s \cup \boldsymbol{w}_i^{us}$, where $\boldsymbol{w}_i^s$ is the salient part and $\boldsymbol{w}_i^{us}$ represents the non-salient elements within group $i$. To effectively keep the information of local salient weights, SQC first introduces the calibration parameter $\tau$ to the SQC quantizer, liberating the perception interval during quantization. Then we define the local salience awareness loss of the SQC quantizer through calibration:

$$\operatorname*{argmin}_{\tau} \ ||\boldsymbol{w}_i^s - \tau \cdot s\{\mathcal{Q}(\boldsymbol{w}_i^s, \tau \cdot s, \tau \cdot z) - \tau \cdot z\}||_2^2 + ||\boldsymbol{w}_i^{us} - \tau \cdot s(\mathcal{Q}(\boldsymbol{w}_i^{us}, \tau \cdot s, \tau \cdot z) - \tau \cdot z)||_2^2 \ (5)$$

where $\mathcal{Q}(\cdot)$ denotes the quantization process in Eq. (1), $|| \cdot ||_2^2$ represents the $\ell_2$ loss, aligned with Eq. (3). $\boldsymbol{w}_i^s$ and $\boldsymbol{w}_i^{us}$ denotes the salient and less salient part of group $i$, respectively, generated from a mask operation. In Eq. (5), $\tau$ expands the solution space of $s$ and $z$, flexibly adjusts $s$ and $z$ to search the optimal loss under $\tau^*$, without bringing additional parameters, as $\boldsymbol{w}_i^s$ and $\boldsymbol{w}_i^{us}$ share the same quantizer. The search space for $\tau$ by linearly dividing the interval [1-$\lambda$, 1+$\lambda$] into $2n$ candidates. We empirically set $\lambda$ at 0.1 and $n$ at 50 to achieve a balance between efficiency and accuracy.

Compared to traditional quantizer calibration methods, SQC effectively mitigates the degradation of intra-group local salient weights caused by general average loss by enhancing the loss sensitivity to salient elements during the calibration (more experiments are detailed in Appendix E). Moreover, the SQC process allows $\boldsymbol{w}_i^s$ and $\boldsymbol{w}_i^{us}$ to share a set of parameters $\tau^*s$ and $\tau^*z$, eliminating the need to differentiate intra-group weights during storage and inference. This facilitates straightforward group-wise dequantization calculations, thereby avoiding the hardware overhead associated with element-wise bitmap and unstructured grouping. SQC and SBA each capture local salient weight information within groups and global salient weight combinations across groups, effectively enhancing the protection of critical information during quantization, thereby accurately preserving the overall performance of LLMs at extremely low bit-widths.

### 3.4 IMPLEMENTATION PIPELINE OF SLIM-LLM

We integrate our mixed-precision framework into advanced PTQ methods, such as GPTQ (Frantar et al., 2022) and OmniQuant (Shao et al., 2023), all of which are inference-friendly with group-wise quantization. We primarily integrate SBA and SQC into GPTQ to get SliM-LLM. For SliM-LLM$^+$, the SBA is plugged into OmniQuant with a learnable quantizer. The plugging pipeline of SliM-LLM is provided in Algorithm 1 (line 4 and line 9), detailed functions are shown in Appendix B.1.

---

**Algorithm 1** Main Framework of SliM-LLM.

---

func SliM-LLM($\boldsymbol{w}$, $\boldsymbol{x}_F$, $\beta$, $\lambda$, $N$)
**Input:** $\boldsymbol{w} \in \mathbb{R}^{n \times m}$ - FP16 weight
    $\boldsymbol{x}_F \in \mathbb{R}^{t \times m}$ - calibration data
    $\beta$ - group size
    $\lambda$ - hessian regularizer
    $N$ - average bit-width
**Output:** $\hat{\boldsymbol{w}}_q$ - quantized weight

1: $\boldsymbol{H} := \frac{1}{P} \sum_{k=1}^{P} \boldsymbol{x}_F^{[k]} \boldsymbol{x}_F^{[k]T}$ hessian matrix
2: $\boldsymbol{H}^{\text{in}} := \text{Cholesky}((\boldsymbol{H} + \lambda \boldsymbol{I})^{-1})$
3: $\hat{\boldsymbol{w}}_q := 0^{n \times m}$

4: $\mathcal{G}\{\cdot\} := \text{SBA}(\boldsymbol{w}, \boldsymbol{x}_F, \boldsymbol{H}^{\text{in}}, \beta, N)$
5: **for** $b = 0, \beta, 2\beta, \dots$ **do**
6: $\quad \boldsymbol{w}^b := \boldsymbol{w}_{:,b:b+\beta}$
7: $\quad g_b := \mathcal{G}[b]$
8: $\quad \boldsymbol{w}_s^b, \boldsymbol{w}_{us}^b := \text{sal\_mask}(\boldsymbol{w}^b)$
9: $\quad \hat{\boldsymbol{w}}_q^b := \text{SQC}(\boldsymbol{w}_s^b, \boldsymbol{w}_{us}^b, g_b)$
10: $\quad$ *GPTQ-error compensation:*
11: $\quad \boldsymbol{E} := (\boldsymbol{w}_{:,b:b+\beta} - \hat{\boldsymbol{w}}_q^b)/\boldsymbol{H}_{bb:b+\beta b+\beta}^{\text{in}}$
12: $\quad \boldsymbol{w}_{:,b+\beta:} := \boldsymbol{w}_{:,b+\beta:} - \boldsymbol{E} \cdot \boldsymbol{H}_{b:b+\beta,b+\beta:}^{\text{in}}$
13: **end for**
14: **return** $\hat{\boldsymbol{w}}_q$

---

Table 1: Quantization results of LLaMA family with statistic quantizer. We report the WikiText2 perplexity in this table, C4 results are shown in Appendix H. '-' denotes that the selected works did not give the results on listed models or the codes

| #W PPL↓ | Method | 1-7B | 1-13B | 1-30B | 1-65B | 2-7B | 2-13B | 2-70B | 3-8B | 3-70B |
|---------|--------|------|-------|-------|-------|------|-------|-------|------|-------|
| 16-bit | - | 5.68 | 5.09 | 4.10 | 3.53 | 5.47 | 4.88 | 3.31 | 5.75 | 2.9 |
| 3-bit | APTQ | 6.76 | - | - | - | - | - | - | - | - |
| | LLM-MQ | - | - | - | - | - | 8.54 | - | - | - |
| | RTN | 7.01 | 5.88 | 4.87 | 4.24 | 6.66 | 5.51 | 3.97 | 27.91 | 11.84 |
| | AWQ | 6.46 | 5.51 | 4.63 | 3.99 | 6.24 | 5.32 | - | 8.22 | 4.81 |
| | GPTQ | 6.55 | 5.62 | 4.80 | 4.17 | 6.29 | 5.42 | 3.85 | 8.19 | 5.22 |
| | **SliM-LLM** | **6.40** | **5.48** | **4.61** | **3.99** | **6.24** | **5.26** | **3.67** | **7.16** | **4.08** |
| 2-bit | LLM-MQ | - | - | - | - | - | 12.17 | - | - | - |
| | RTN | 1.9e3 | 781.20 | 68.04 | 15.08 | 4.2e3 | 122.08 | 27.27 | 1.9e3 | 4.6e5 |
| | AWQ | 2.6e5 | 2.8e5 | 2.4e5 | 7.4e4 | 2.2e5 | 1.2e5 | - | 1.7e6 | 1.7e6 |
| | GPTQ | 152.31 | 20.44 | 13.01 | 9.51 | 60.45 | 28.14 | 8.78 | 210.00 | 11.90 |
| | QuIP | 29.74 | 12.48 | 11.57 | 7.83 | 39.73 | 13.48 | 6.64 | 84.97 | 13.03 |
| | PB-LLM | 24.61 | 17.73 | 12.65 | 7.85 | 25.37 | 49.81 | NAN | 44.12 | 11.68 |
| | **SliM-LLM** | **14.58** | **8.87** | **7.33** | **5.90** | **16.01** | **9.41** | **6.28** | **39.66** | **9.46** |

Table 2: Quantization results of LLaMA-1 and LLaMA-2 models with learnable quantizer. We report the WikiText2 perplexity in this Table, C4 results are shown in Appendix H. '-' denotes that the selected works have not reported the results on listed models or published the codes

| #W PPL↓ | Method | 1-7B | 1-13B | 1-30B | 1-65B | 2-7B | 2-13B | 2-70B |
|---------|--------|------|-------|-------|-------|------|-------|-------|
| 16-bit | - | 5.68 | 5.09 | 4.10 | 3.53 | 5.47 | 4.88 | 3.31 |
| 3-bit | OmniQuant | 6.15 | 5.44 | 4.56 | 3.94 | 6.03 | 5.28 | 3.78 |
| | AffineQuant | 6.14 | 5.45 | 4.59 | - | 6.08 | 5.28 | - |
| | **SliM-LLM$^+$** | **6.07** | **5.37** | **4.34** | **3.72** | **5.94** | **5.11** | **3.35** |
| 2-bit | OmniQuant | 9.72 | 7.93 | 7.12 | 5.95 | 11.06 | 8.26 | 6.55 |
| | AffineQuant | 13.51 | 7.22 | 6.49 | - | 10.87 | 7.64 | - |
| | **SliM-LLM$^+$** | **9.68** | **7.17** | **6.41** | **5.74** | **10.87** | **7.59** | **6.44** |

## 4 EXPERIMENTS

We evaluated SliM-LLM and SliM-LLM$^+$ under weight-only conditions, focusing on 2/3-bit precisions. Per-channel group quantization is utilized in our framework with $groupsize = 128$ in experiments. Since no back-propagation in SliM-LLM, the quantization is carried out on a single NVIDIA A800 GPU. For SliM-LLM$^+$, we employ the AdamW optimizer, following OmniQuant (Shao et al., 2023), which is also feasible on a single A800. We randomly select 128 samples from WikiText2 (Merity et al., 2016) as calibration data, each with 2048 tokens.

**Models and Evaluation.** To comprehensively demonstrate the low-bit performance advantages of SliM-LLM and SliM-LLM$^+$, we conduct experiments across OPT (Zhang et al., 2022), LLaMA (Touvron et al., 2023a), LLaMA-2 (Touvron et al., 2023b) and LLaMA-3. We employ the perplexity as our evaluation metric, which is widely recognized as a stable measure of language generation capabilities (Frantar et al., 2022; Lin et al., 2023; Huang et al., 2024a; Shang et al., 2023; Shao et al., 2023; Chee et al., 2024; Egiazarian et al., 2024; Huang et al., 2024b), particularly in compression scenarios. Experiments are carried out on the WikiText2 (Merity et al., 2016) and C4 (Raffel et al., 2020)datasets. Furthermore, to assess the practical application capabilities of quantized LLMs, we also evaluate their accuracy on zero-shot benchmarks, including PIQA (Bisk et al., 2020), ARC (Clark et al., 2018), BoolQ (Clark et al., 2019), and HellaSwag (Clark et al., 2018).

**Baseline.** Since SliM-LLM and SliM-LLM$^+$ are efficient PTQ approaches without additional training or fine-tuning, QAT and re-training methods are not within the comparison range of our work. The experiments evaluate existing advanced quantization methods and GPU-friendly computations, including vanilla round-to-nearest (RTN), GPTQ (Frantar et al., 2022), AWQ (Lin et al., 2023). And mixed-precision quantization techniques, including PB-LLM (Shang et al., 2023) ($\frac{1}{7} \times$ 8-bit+$\frac{6}{7} \times$ 1-bit), LLM-MQ (Li et al., 2024a), and APTQ (Guan et al., 2024), as well as the codebook-based method QuIP (Chee et al., 2024) are also compared in this work. We compare SliM-LLM$^+$ with

Table 3: Performance comparisons of different quantization methods for zero-shot tasks.

| Model / Acc↑ | #W | Method | PIQA | ARC-e | ARC-c | BoolQ | HellaSwag | Winogrande | Avg. |
|---|---|---|---|---|---|---|---|---|---|
| LLaMA-7B | 16-bit | - | 77.47 | 52.48 | 41.46 | 73.08 | 73.00 | 67.07 | 64.09 |
| | 2-bit | GPTQ | 55.49 | 31.02 | 22.17 | 53.49 | 33.84 | 41.91 | 39.65 |
| | 2-bit | AWQ | 47.78 | 28.77 | 21.31 | 31.19 | 24.47 | 40.03 | 32.26 |
| | 2-bit | **SliM-LLM** | **57.83** | **33.46** | **25.09** | **56.05** | **36.70** | **52.64** | **43.84** |
| | 2-bit | OmniQuant | 63.63 | 43.91 | 27.32 | 58.02 | 48.78 | 52.97 | 49.11 |
| | 2-bit | **SliM-LLM$^+$** | **64.96** | **45.66** | **28.67** | **64.59** | **48.86** | **53.35** | **51.02** |
| LLaMA-13B | 16-bit | - | 79.10 | 59.89 | 44.45 | 68.01 | 76.21 | 70.31 | 66.33 |
| | 2-bit | GPTQ | 70.37 | 47.74 | 35.88 | 51.57 | 61.39 | 60.84 | 54.63 |
| | 2-bit | AWQ | 49.23 | 30.01 | 29.49 | 30.88 | 26.72 | 46.30 | 35.44 |
| | 2-bit | **SliM-LLM** | **73.19** | **47.95** | **36.27** | **55.92** | **63.04** | **61.79** | **56.36** |
| | 2-bit | OmniQuant | 73.14 | 49.38 | 36.93 | 63.34 | 62.19 | 61.77 | 57.64 |
| | 2-bit | **SliM-LLM$^+$** | **74.15** | **50.26** | **37.04** | **64.31** | **63.57** | **63.11** | **58.74** |
| LLaMA-30B | 16-bit | - | 80.08 | 58.92 | 45.47 | 68.44 | 79.21 | 72.53 | 67.44 |
| | 2-bit | GPTQ | 71.92 | 48.27 | 36.20 | 61.27 | 65.76 | 63.11 | 57.76 |
| | 2-bit | AWQ | 49.17 | 28.56 | 25.97 | 34.73 | 24.97 | 46.99 | 35.07 |
| | 2-bit | **SliM-LLM** | **75.52** | **51.29** | **39.29** | **62.01** | **66.10** | **64.07** | **59.71** |
| | 2-bit | OmniQuant | 76.23 | 53.23 | 39.52 | 63.34 | 65.57 | 64.82 | 60.22 |
| | 2-bit | **SliM-LLM$^+$** | **76.31** | **54.07** | **39.79** | **63.35** | **67.14** | **64.93** | **60.91** |
| LLaMA-65B | 16-bit | - | 80.79 | 58.71 | 46.24 | 82.29 | 80.72 | 77.50 | 71.04 |
| | 2-bit | GPTQ | 76.16 | 52.48 | 40.14 | 77.23 | 71.96 | 70.22 | 64.70 |
| | 2-bit | **SliM-LLM** | **77.09** | **53.72** | **40.25** | **77.51** | **72.05** | **70.91** | **65.26** |
| | 2-bit | OmniQuant | 77.78 | 53.71 | 40.90 | 78.04 | 74.55 | 68.85 | 65.64 |
| | 2-bit | **SliM-LLM$^+$** | **78.06** | **53.90** | **41.18** | **78.33** | **75.59** | **69.99** | **66.18** |

gradient optimizer-based OmniQuant (Shao et al., 2023) and AffineQuant (Ma et al., 2024b). When applying SliM-LLM, the quantization process for a 7B model takes only about 50 minutes.

## 4.1 MAIN RESULTS

We show experiments within the LLaMA family in this section and detailed results for the OPT models are available in Appendix H. For language generation tasks, as depicted in Tab. 1, SliM-LLM markedly outperforms its backbone GPTQ, particularly under the 2-bit. Specifically, on LLaMA-7B, SliM-LLM achieves a 90% decrease in perplexity, while on LLaMA-3-8B, it improves by 81%. In comparison with the element-wise mixed-precision PB-LLM and the codebook-based QuIP method, SliM-LLM further reduces the perplexity by 41%~51%. As shown in Tab. 1, the performance of SliM-LLM$^+$ is still ahead compared to OmniQuant and AffineQuant, further proving the effectiveness and of the mixed-precision framework. We also provide dialogue examples of 2-bit instruction fine-tuning Vicuna-13B (Chiang et al., 2023) and LLaMA-13B in Appeandix I.

Morever, our method exhibits zero-shot advantages at 2-bit, as shown in Tab. 3, where SliM-LLM and SliM-LLM$^+$ still outperforms other methods. For instance, compared with GPTQ and OmniQuant, our approach achieves an average improvement of 4.19% and 1.91% on LLaMA-7B. Meanwhile, for LLaMA-65B, 2-bit SliM-LLM and SliM-LLM$^+$ is close to FP16 results (less than 6% degradaion in accuracy). Overall, our proposed mixed-precision framwork demonstrates superior performance across different model sizes, with its advantages becoming increasingly significant at lower bit-width.

## 4.2 ABLATION RESULTS

**Abliation of SBA and SQC**. We conduct a detailed ablation study to illustrate the benefits of bit-width allocation and the impact of each component. Fig. 5(a) compares three strategies for allocating bit-widths across groups, including random allocation, head-tail allocation by spatial order, and our proposed SBA. When the

Table 4: WikiText2↓ performance of SBA and ILP on LLaMA.

| Method | #W | 7B | 13B | 30B | 65B |
|---|---|---|---|---|---|
| ILP | 2-bit | 17.55 | 9.51 | 9.27 | 7.46 |
| **SBA** | 2-bit | **14.58** | **8.87** | **7.33** | **5.90** |

average bit-width remains constant, random and head-tail mixed-precision allocation prove ineffective and even result in performance degradation, as shown in Fig. 5(a). In contrast, SBA consistently delivers significant improvements in post-quantization performance, validating the efficacy of our

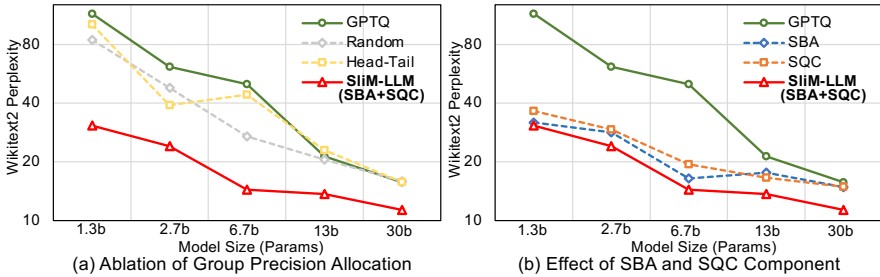

Figure 5: Ablation results on OPT models. Random means randomly selecting the same number of lower/higher-bit groups; head-tail denotes using the head groups as the lower-bit and the same number of tails as the higher-bit on the original sequence of group.

Table 5: Deployment results of GPTQ and Slim-LLM on GPU. Group size is set to 128.

| #W | LLaMA-* | 1-7B | | | | 1-13B | | | | 2-7B | | | |
|---|---|---|---|---|---|---|---|---|---|---|---|---|---|
| | | WM | RM | PPL↓ | Token/s | WM | RM | PPL↓ | Token/s | WM | RM | PPL↓ | Token/s |
| FP16 | - | 12.6G | 14.4G | 5.68 | 69.2 | 24.3G | 27.1G | 5.09 | 52.5 | 12.7G | 14.6G | 5.47 | 69.3 |
| 3-bit | GPTQ | 3.2G | 5.1G | 6.55 | 83.4 | 5.8G | 8.7G | 5.62 | 57.6 | 3.2G | 5.2G | 6.29 | 56.3 |
| | **SliM-LLM** | 3.2G | 5.2G | **6.40** | 79.1 | 5.8G | 8.8G | **5.48** | 48.5 | 3.2G | 5.4G | **6.26** | 55.9 |
| 2-bit | GPTQ | 2.2G | 4.1G | 152.31 | 83.9 | 4.0G | 7.5G | 20.44 | 92.6 | 2.2G | 4.1G | 60.45 | 83.6 |
| | **SliM-LLM** | 2.3G | 4.4G | **14.58** | 61.2 | 4.1G | 7.8G | **8.87** | 73.7 | 2.3G | 4.1G | **16.01** | 64.4 |

mixed-precision approach. Fig. 5(b) presents the ablation effects of SBA and SQC, demonstrating that both methods, based on the perception of global and local salience, enhance quantization performance. SBA is particularly effective in smaller models, and combining these two methods can further boost capabilities of LLMs. We also provide the detailed ablation results on group size in Appendix F.

**Compare of SBA and ILP**. We compare the performance between the ILP model in HAWQ v2 (Dong et al., 2019) and SBA on the LLaMA model. Tab. 4 shows that SBA achieves comprehensive performance superiority on LLaMA. We observed that under a 2-bit scenario, ILP ensures an equal number of 1-bit and 3-bit groups within the search space {1-bit, 2-bit, 3-bit}. The advantage of ILP lies in a broader selection range for target bit-widths, but under commonly used fixed integer bit-widths (e.g. 2-bit, 3-bit), SBA's double-pointer search strategy based on output feature KL proposed by SBA can achieve a more optimal matching strategy.

### 4.3 EFFICIENT INFERENCE ON DEVICE

We utilize the open-source AutoGPTQ to extend CUDA kernel supporting experimental mixed-precision inference, with detailed process in Appendix B.2. We evaluate the deployment performance of LLaMA-7/13B and LLaMA-2-7B under 2/3-bit settings in Tab. 5. The results indicate that our mixed-precision approach maintains a good compression rate on GPUs and significantly enhances model accuracy, only with a slight decrease in inference speed on the A800 (due to the inference alignment of different bit-width). Since current 1-bit operations lack well hardware support, additional consumption of storage and computation is required on device. There remains considerable scope for optimization in mixed-precision computing, and we aim to further improve this in future work.

### 5 CONCLUSION

In this work, we introduce **SliM-LLM**, a group-wise mixed-precision PTQ framework tailored for LLMs, designed to enhance performance with low-bit weights in a deployment-friendly manner. The essence of SliM-LLM lies in employing the *Salience-Determined Bit Allocation* to dynamically allocate bit widths, thereby improving the preservation of global salience information. Within groups, the *Salience-Weighted Quantizer Calibration* is designed to enhance local information perception, further minimizing the loss associated with locally salient weights. Experiments validate the effectiveness of SliM-LLM, showing notable accuracy improvements across various LLMs, and ensuring efficiency in inference. In conclusion, SliM-LLM is versatile and can be seamlessly integrated with different quantization frameworks and successfully improves the performance of LLMs supporting practical deployment in resource-constrained environments.

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

# A  LIMITATIONS

Though the mixed-precision framework significantly improves the quantization performance of LLMs, the current out-of-the-box deployment tools still cannot well support efficient mixed-precision computing. Meanwhile, the support for 1/2/3-bit inference on GPUs remains limited, which affects the inferencing advantages of low-bit models. We believe there is significant room for improvement in the hardware efficiency of mixed-precision LLMs in the future.

# B  SLIM-LLM IMPLEMENTATION

## B.1  DETAILED IMPLEMENTATION

In this section, we present the specific implementation details of SliM-LLM, which utilizes GPTQ (Frantar et al., 2022) as its backbone for mixed-precision quantization and incorporates both SBA and SQC. SliM-LLM$^+$ is consistent with SliM-LLM in SBA computations but does not include the SQC component, instead retaining learnable weight clipping (LWC) approach in OmniQuant (Shao et al., 2023) for gradient optimization.

---

**Algorithm 2** Detailed functions in SliM-LLM.

func $\text{SBA}(\boldsymbol{w}, \boldsymbol{x}_F, \boldsymbol{H}^{\text{in}}, \beta, N)$

1: $\mathcal{G}\{\cdot\} := \{0\}$ // initialize group bit-width
2: $e := \inf$ // bit-width searching error
3: $p^* := 0$ // number of $(N{-}1)$-bit and $(N{+}1)$-bit
4: $l := N - 1$ // lower bit-width
5: $h := N + 1$ // higher bit-width
6: $S\{\cdot\} := \text{average}(\frac{\boldsymbol{w}^2}{[\boldsymbol{H}^{\text{in}}]^2_{\text{diag}}})$
7: **for** $p = 1, 2, ..., [\frac{m}{2\beta}]$ **do**
8: $\quad \hat{\boldsymbol{w}}_l^b := \text{fakequant}(\boldsymbol{w}_{b\in\text{top\_k\_min}(p)}^b, l, )$
9: $\quad \hat{\boldsymbol{w}}_h^b := \text{fakequant}(\boldsymbol{w}_{b\in\text{top\_k\_max}(p)}^b, h, )$
10: $\quad \hat{\boldsymbol{w}}_N^b := \text{fakequant}(\boldsymbol{w}_{b\in\text{others}}^b, N, )$
11: $\quad \hat{\boldsymbol{w}}_q := \hat{\boldsymbol{w}}_l^b \cup \hat{\boldsymbol{w}}_l^b \cup \hat{\boldsymbol{w}}_h^b$
12: $\quad$ **if** $\mathcal{D}_{kl}(\boldsymbol{x}\boldsymbol{w}^\top \| \boldsymbol{x}\hat{\boldsymbol{w}}_q^\top) < e$ **then**
13: $\quad\quad e := \mathcal{D}_{kl}(\boldsymbol{x}\boldsymbol{w}^\top \| \boldsymbol{x}\hat{\boldsymbol{w}}_q^\top)$
14: $\quad\quad p^* := p$
15: $\quad$ **end if**
16: **end for**
17: $\mathcal{G}\{l\} := S\{\text{top\_k\_min}(p^*) = l\}$
18: $\mathcal{G}\{h\} := S\{\text{top\_k\_max}(p^*) = h\}$
19: $\mathcal{G}\{N\} := S\{\text{middle\_k}([\frac{m}{2}] - 2p^*) = N\}$
20: **return** $\mathcal{G}\{\cdot\}$

func $\text{SQC}(\boldsymbol{w}_s^b, \boldsymbol{w}_{us}^b, g_b)$

1: $w_{\max} := \max(\boldsymbol{w}_s^b \cup \boldsymbol{w}_{us}^b)$
2: $w_{\min} := \min(\boldsymbol{w}_s^b \cup \boldsymbol{w}_{us}^b)$
3: $\lambda := 0.1$
4: $n := 50$
5: $e := \inf$ // scale searching error
6: $\Delta^* \in \mathbb{R}^{n\times 1}$ // per-channel scale
7: $z^* \in \mathbb{R}^{n\times 1}$ // per-channel zero point
8: **for** $\tau \in [1 - \lambda, 1 + \lambda]$ with $2n$ slices **do**
9: $\quad \Delta := \tau(w_{\max} - w_{\min})/(2^{g_s} - 1)$
10: $\quad z := -\lfloor (\tau w_{\min})/\Delta \rfloor$
11: $\quad \hat{\boldsymbol{w}}_s^b := \text{fakequant}(\boldsymbol{w}_s^b, g_b, \Delta, z)$
12: $\quad \hat{\boldsymbol{w}}_{us}^b := \text{fakequant}(\boldsymbol{w}_{us}^b, g_b, \Delta, z)$
13: $\quad \mathcal{L}_s := \|\boldsymbol{w}_s^b - \hat{\boldsymbol{w}}_s^b\|^2$
14: $\quad \mathcal{L}_{us} := \|\boldsymbol{w}_{us}^b - \hat{\boldsymbol{w}}_{us}^b\|^2$
15: $\quad$ **if** $\mathcal{L}_s + \mathcal{L}_{us} < e$ **then**
16: $\quad\quad e := \mathcal{L}_s + \mathcal{L}_{us}$
17: $\quad\quad z^* := z$
18: $\quad\quad \Delta^* := \Delta$
19: $\quad$ **end if**
20: **end for**
21: $\hat{\boldsymbol{w}}_q^b := \text{fakequant}(\boldsymbol{w}^b, g_b, \Delta^*, z^*)$
22: **return** $\hat{\boldsymbol{w}}_q^b$

---

Algorithm 2 primarily encompasses the core details of both SBA and SQC. In SBA, the importance of each group is determined by sorting the average salience of groups, followed by a bi-pointer search that increases the number of $(N - 1)$-bit and $(N + 1)$-bit groups to maintain their quantity equilibrium. The optimization function then utilizes the KL divergence from Eq. (4) to determine the optimal mixed-precision ratio. SQC, on the other hand, enhances its information by amplifying the quantization error of unstructured weight groups. When the last two parameters, scale and zero point, in the $\text{fakequant}(\cdot)$ function are omitted, the default values from Eq. (1) are used.

## B.2  MIXED BIT STORAGE AND COMPUTING

We developed a framework for storage and inference deployment supporting mixed-precision quantization based on AutoGPTQ. The deployment process is as follows. After completing mixed-precision

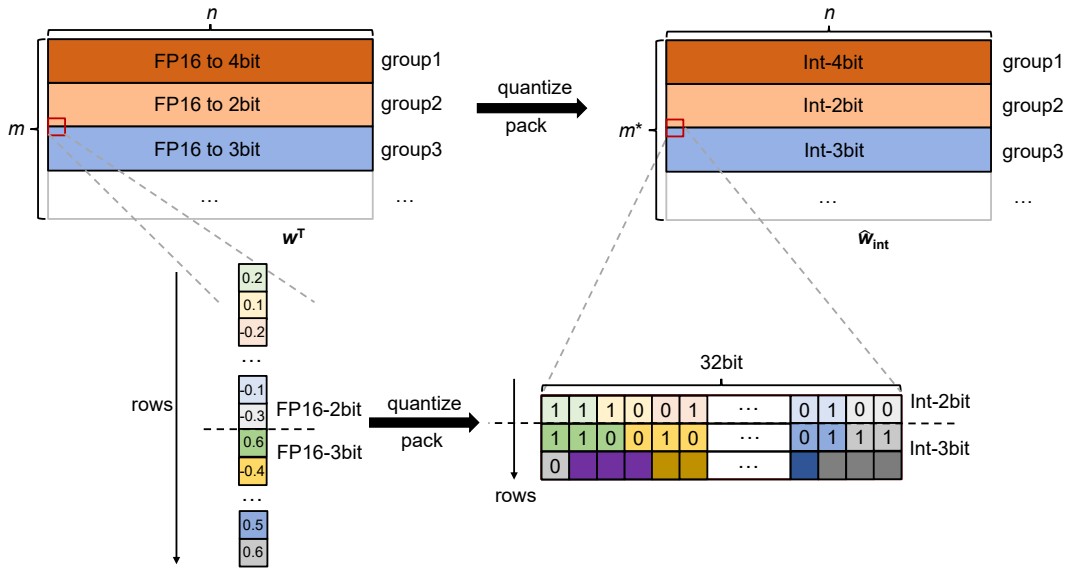

Figure 6: The memory layout shown in the figure is modified based on AutoGPTQ. The transposed original weights $\boldsymbol{w}^\top \in \mathbb{R}^{m \times n}$ are still divided into multiple groups along the row direction after quantization. The elements within each group are vertically packed into integers and then reassembled into $\hat{\boldsymbol{w}}_{\text{int}}$. The figure employs corresponding colors to indicate how each original number is mapped to a specific position within the packed integers after quantization, which finally generates $\hat{\boldsymbol{w}}_{\text{int}} \in \mathbb{R}^{m^* \times n}$, where $m^*$ is compressed from $m$ by packing several low-bit number. Similarly, $\hat{\boldsymbol{z}}_{\text{int}}$ is also packed into integers to save memory.

quantization with SliM-LLM, it outputs scales, zeros, and group-wise bit-width generated during the quantization process to identify the quantization parameters and precision of each group in the Linear Projection weights. AutoGPTQ then packs the weights and zeros into integer-compressed representations (denoted by $\hat{\boldsymbol{w}}_{\text{int}}$ and $\hat{\boldsymbol{z}}_{\text{int}}$ respectively) based on the precision of different groups, significantly reducing storage and operational bit-width. After the quantized weights are packed, AutoGPTQ loads the model onto the GPU, where the mixed precision quantization kernel on the GPU performs dequantization on the weights and zeros of different groups and calculation with input activation, ultimately producing the final output.

In the mixed-precision deployment of AutoGPTQ, the weight memory layout is organized by group, with each group sharing the same precision, which is shown in Fig. 6. Within each group, elements with the same precision are packed as integers, eliminating the need for additional padding, which saves space. Given that the bit-width of integers is a power of 2, this is compatible with group size that is also a power of 2. For instance, even with the odd-bit such as 3-bit storage, integers can store these numbers without padding, as the commonly used group size is 128, a multiple of almost all definition of integer type. This ensures that elements within a group fully utilize the space provided by integers, without storing numbers of different precision within the same integer. $\hat{\boldsymbol{z}}_{\text{int}}$ follow the original logic of AutoGPTQ but are packed with a uniform precision along the channel direction for ease of use. Other tensors, like scales, remain in the same floating-point format to ensure the correctness of dequantization calculations.

To indicate the precision of each group, we also introduce an additional array to store bit-width of each group, where each number is represented as a 2-bit value aggregated into integers, marking the quantization precision of each group for accurate reconstruction. We use cumulative calculations to determine the starting index of each group, ensuring correctness despite changes in $\hat{\boldsymbol{w}}_{\text{int}}$ height and starting indices caused by varying precision. Using the above methods to store the quantized weights, zeros, and additional bit arrays effectively reduces memory usage during model storage and loading, thereby lowering the resource overhead required for model deployment.

Once the weights are packed, we follow the modified AutoGPTQ logic for GPU inference. The GPU processes and dequantizes the weights group by group for computation. During GPU computation,

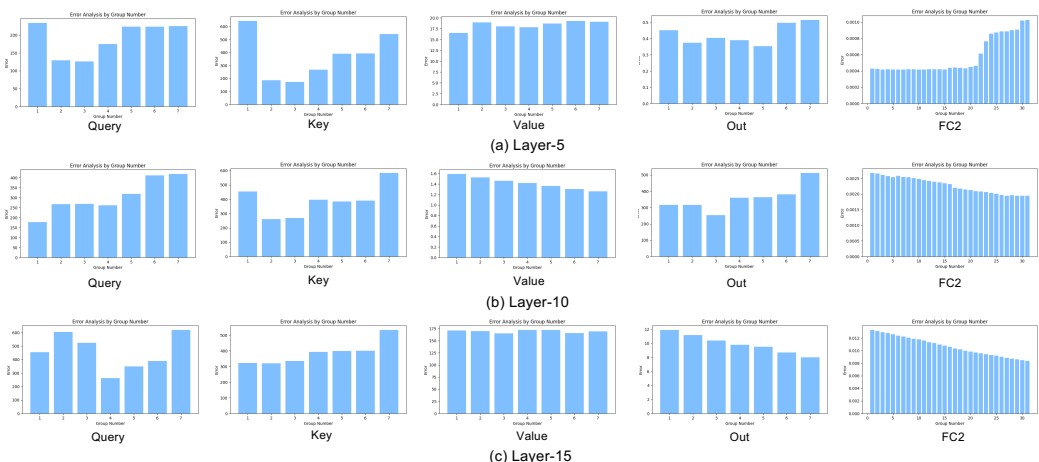

Figure 7: Error curves of SBA for select weights in the $5^{th}$, $10^{th}$, and $15^{th}$ layers of OPT-1.3B.

a thread dequantizes a segment of continuous memory data in one column of $\hat{w}_{\text{int}}$ and performs vector dot product calculations with the input activation shared within the block, accumulating the results in the corresponding result matrix. When threads form a logical block, the block handles the computation and reduction of a continuous channel region. We complete the linear layer computation by iterating through all logical blocks. Leveraging AutoGPTQ's initial logic and CUDA Warp's 32-thread units, we ensure similar code structure and data access logic for threads within each warp when group size is 128. This method was primarily conducted to validate feasibility os SliM-LLM, demonstrating that the mixed precision quantization with integer packing does not cause additional computational overhead, indicating the efficiency and accuracy advantage of SliM-LLM. In summary, by dividing weight into several structured groups with mixed precision and employing a reasonable GPU utilization strategy, Slim-LLM balances performance and efficiency.

## C    SEARCHING DETAILS OF GROUP-WISE SALIENCE-DETERMINED BIT ALLOCATION

We optimize the mixed-precision configuration based on the output information entropy (KL-divergence), searching for the optimal compensation bit-width ratio as shown in Eq. (4).

Initially, we rank each group by their average salience, a metric for quantization, and employ a double-pointer that moves simultaneously from both the beginning (lowest salience) and end (highest salience) of the sorted list. This ensures an equal number of groups at low and high bit-widths, effectively balancing the global average bit-width compensation. We then calculate the relative entropy under the corresponding precision ratio and search for the optimal ratio. Fig 7 displays the search error curves related to the $2^{nd}$, $10^{th}$, and $15^{th}$ Transformer layers in the OPT1.3B model, showcasing the search curves for certain self-attention layers (Query, Key, Value, FC2).

Due to the limited range of the search, extreme scenarios involve either a half $(N-1)$-bit and half $(N+1)$-bit without $N$-bit or all groups being $N$-bit (uniform precision). Fig 7 demonstrates that lower quantization errors can be achieved under mixed-precision compared to quantization at the uniform bit-width. We also find that multiple low-error precision combinations are possible within a group of weights, allowing SBA to flexibly select the optimal ratio through its versatile search.

## D    EVLUATIION FUNCTION OF SBA

In Tab. 6, we employ various objective functions and compare their performance in SBA across different models. Compared to the commonly used Mean Squared Error (MSE) loss, Kullback-Leibler (KL) divergence ensures the distribution of critical activation positions within the model from an information entropy perspective, making it a superior choice for the bit-width allocation strategy in

SBA for the OPT and LLaMA models. When computing KL divergence in this context, we first transform the layer outputs into probability distributions using softmax.

Table 6: Comparison of MSE and KL Divergence in SBA.

| Method | # W | OPT-1.3B | OPT-2.7B | OPT-6.7B | OPT-13B | LLaMA-7B | LLaMA2-7B |
|---|---|---|---|---|---|---|---|
| MSE | 2-bit | 32.50 | 27.58 | 15.14 | 13.28 | 21.94 | 16.86 |
| **KL Divergence** | 2-bit | **30.71** | **13.26** | **11.27** | **10.12** | **14.58** | **16.01** |

# E  EXTENSION ABLATION ON SQC

In this section, we visualize the effectiveness of SQC in mitigating the degradation of information in locally salient weights. We observed the absolute error of weights in a randomly selected channel of the quantized OPT-1.3B model. As shown in Fig. 8, the overall absolute error of the weights post-quantization with a standard quantizer was 0.0055, while with SQC it was reduced to 0.0039. This further demonstrates that the search parameter $\tau$, as applied in Eq. (5), effectively optimizes the quantizer parameters, thereby reducing quantization errors.

More importantly, SQC effectively perceives the information of locally salient weights, as indicated by the red regions in Fig. 8. Compared to the vanilla quantizer, SQC significantly reduces the error of salient weights. Specifically, the prominent weights at indices 375 in Fig. 8(a) show higher quantization errors, while in Fig. 8(b), this error is effectively reduced. This confirms SQC's ability to perceive locally salient weights, effectively preventing the degradation of critical information.

# F  EXTENSION ABLATION ON QUANTIZATION GROUP-SIZE

To investigate the impact of different group sizes on the quantization effectiveness of SliM-LLM, we evaluated performance with 256 and 512 columns at a 3-bit level, observing that larger group sizes enhance GPU efficiency during inference. The findings suggest that increased group granularity does not substantially elevate perplexity across four models, indicating that SliM-LLM is robust and conducive to more efficient deployment methods. In contrast, at 2-bit, we assessed group sizes of 64 and 32 columns. With finer group granularity, the models displayed reduced perplexity. This is attributed to smaller groups providing more detailed data representation and utilizing additional quantization parameters, although they also raise computational and storage demands. A group size of 128 strikes a better balance between efficiency and quantization performance.

# G  EXTENSION ON SALIENCE CHANNEL CLUSTERING

## G.1  DISCUSSION OF THEOREM 1

**Theorem 1.** *Given the input calibration activation $\boldsymbol{x} \in \mathbb{R}^{t \times m}$ with an outlier channel $\boldsymbol{x}^*_{:,p} \gg \boldsymbol{x}_{:,j}, \forall j \in [0, m], j \neq p$ at the position of channel-p. The trace elements of $\boldsymbol{H} = \boldsymbol{x}^\top \boldsymbol{x}$ will show great outlier value at $(p, p)$, where $\boldsymbol{H}_{p,p} \gg \boldsymbol{H}_{j,j}, \forall j \in [0, m], j \neq p$, as $\boldsymbol{H}_{p,p}$ is produced by $[\boldsymbol{x}^{*\top}_{:,p} \boldsymbol{x}^*_{:,p}] = \sum_{i=0}^{t} x^{*2}_{i,p}$, which further leads to the parameter salience larger at the $p^{th}$ channel of weight, where $\delta_{:,p} > \delta_{:,k}, \delta_{:,k} = \frac{w^2_{:,k}}{[\boldsymbol{H}^{-1}]^2_{k,k}}, \forall k \in [0, t], k \neq p.$*

*Proof.* Given $\boldsymbol{x} \in \mathbb{R}^{t \times m}$ with outlier channel $\boldsymbol{x}^*_{:,p}$, $p \in [0, m]$, and other elements with small magnitude $x_{i,j}$, where $x^*_{q,p} \gg x_{i,j}$ and $i, j \neq q, p$. We can get the Hessian matrix with Levenberg-

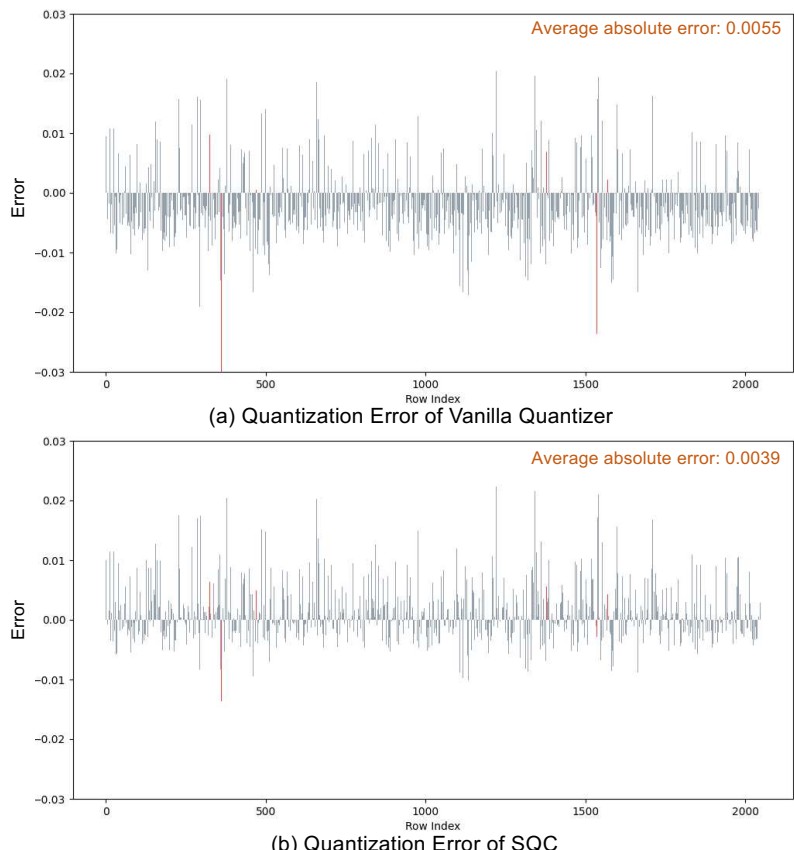

Figure 8: Absolute channel error of the weight of the OPT-1.3B model. The red line represents the quantization error for the locally salient weights, and the lightmauve represents other weights. (a) Vanilla quantizer error on the $794^{th}$ channel of OPT-1.3B. (b) SQC error on the $794^{th}$ channel of OPT-1.3B

Table 7: Ablation results on OPT-6.7B, LLaMA-7B, LLaMA-2-7B, LLaMA-3-8B with SliM-LLM under different group size (#g denotes the group size).

| Precision / PPL↓ | #g | OPT-6.7B | LLaMA-7B | LLaMA-2-7B | LLaMA-3-8B |
|---|---|---|---|---|---|
| | 512 | 11.65 | 6.96 | 6.69 | 8.87 |
| 3-bit | 256 | 11.33 | 6.92 | 6.94 | 8.14 |
| | 128 | 11.27 | 6.40 | 6.24 | 7.62 |
| | 128 | 14.41 | 14.58 | 16.01 | 39.66 |
| 2-bit | 64 | 13.95 | 13.41 | 15.02 | 29.84 |
| | 32 | 12.47 | 11.91 | 11.95 | 16.93 |

Marquardt (Marquardt, 1963) approximation in Eq. (3):

$$
\begin{pmatrix}
x_{11} & x_{12} & x_{13} & \cdots & x_{1m} \\
x_{21} & x_{22} & x_{23} & \cdots & x_{2m} \\
\vdots & \vdots & \vdots & \ddots & \vdots \\
\vdots & \vdots & \vdots & \boldsymbol{x^*_{q,p}} & \vdots \\
\vdots & \vdots & \vdots & \ddots & \vdots \\
x_{t1} & x_{t2} & x_{t3} & \cdots & x_{tm}
\end{pmatrix}
\cdot
\begin{pmatrix}
x_{11} & x_{21} & \cdots & x_{t1} \\
x_{21} & x_{22} & \cdots & x_{t2} \\
\vdots & \vdots & \ddots & \vdots \\
\vdots & \vdots & \boldsymbol{x^*_{q,p}} & \vdots \\
\vdots & \vdots & \ddots & \vdots \\
x_{1m} & x_{2m} & \cdots & x_{tm}
\end{pmatrix}
=
\begin{pmatrix}
x_{11}^2 + .. & \cdots & \cdots & \cdots \\
\vdots & \ddots & \cdots & \vdots \\
\vdots & \vdots & \boldsymbol{x^*_{1,p}}^2 + .. & \vdots \\
\cdots & \cdots & \cdots & \ddots
\end{pmatrix}
\tag{6}
$$

where $[\boldsymbol{x}_{:,\boldsymbol{p}}^{*\top}\boldsymbol{x}_{:,\boldsymbol{p}}^{*}]$ will appears at position $\boldsymbol{H}_{p,p}$. And following SparseGPT (Frantar & Alistarh, 2023), the inverse matrix of $\boldsymbol{H}$ can be formulated as:

$$\delta_{i,j} = \frac{w_{i,j}^2}{[\text{diag}((\boldsymbol{x}^\top \boldsymbol{x} + \lambda \boldsymbol{I})^{-1})]^2} \tag{7}$$

where $(\boldsymbol{x}^\top \boldsymbol{x} + \lambda \boldsymbol{I})^{-1}$ is the new representation of Hessian matrix $\boldsymbol{H}$ for the layer-wise reconstruction problem, and $\lambda$ is the dampening factor for the Hessian to prevent the collapse of the inverse computation. Additionally, in accordance with the configuration in LLMs (Frantar & Alistarh, 2023; Frantar et al., 2022; Sun et al., 2023), the value of $\lambda$ set is extremely small ($\lambda \leq e^{-1}$), while the values located at the diagonal of Hessian are large. Therefore, only considering the influence of diagonal elements (Sun et al., 2023), we can further approximate salience as:

$$\delta_{i,j} = \frac{w_{i,j}^2}{[\text{diag}((\boldsymbol{x}^\top \boldsymbol{x} + \lambda \boldsymbol{I})^{-1})]^2} \approx \frac{w_{i,j}^2}{[(\text{diag}(\boldsymbol{x}^\top \boldsymbol{x}))^{-1}]^2} = (w_{i,j} \cdot ||\boldsymbol{x}_j||_2^2)^2 \tag{8}$$

Here the diagonal of $\boldsymbol{x}^\top \boldsymbol{x}$ is $\text{diag}(||\boldsymbol{x}_j||_2^2)$, and $||\boldsymbol{x}_j||_2$ evaluates the $\ell_2$ norm of $j^{th}$ channel across different tokens. Consequently, it can be summarized that when there is an outlier channel-$p$, the value of $||\boldsymbol{x}_p||_2$ is primarily influenced by $[\boldsymbol{x}_{:,\boldsymbol{p}}^{*\top}\boldsymbol{x}_{:,\boldsymbol{p}}^{*}]$. Additionally, since the activation values are relatively large and the differences in weight values are comparatively small, the $p^{th}$ channel of weights will also exhibit salience. □

### G.2 DISTRIBUTION OF SALIENCE, ACTIVATION AND WEIGHT MAGNITUDE

Fig. 9 illustrates the distribution of salience among certain weights in LLMs. This section provides additional examples to demonstrate how the distribution of weights and input activation characteristics influence the salience of parameters in LLMs. The figure captures seven linear projections in the multi-head self-attention (MHA) and feed-forward block (FFB) layers of the $2^{nd}$ and $10^{th}$ Transformer modules in the LLaMA-7B model.

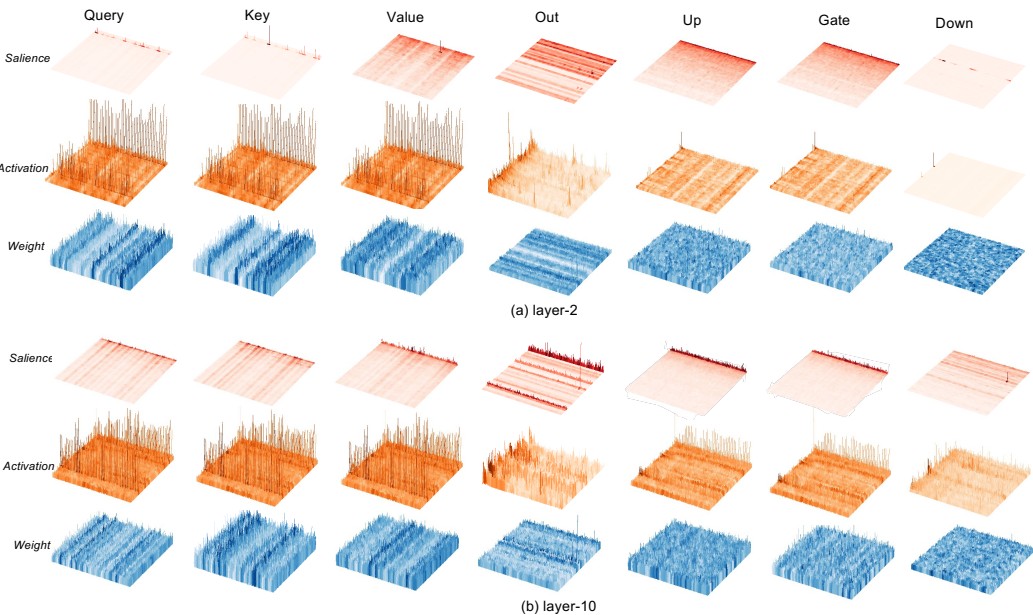

Figure 9: Salience, activation and weight distribution in the $2^{nd}$ and $10^{th}$ layers of LLaMA-7B

In line with previous findings (Nrusimha et al., 2024; Xiao et al., 2023a), activations demonstrate particularly marked outlier phenomena on anomalous tokens and channels, with extremes differing by more than two orders of magnitude. Notably, distinct anomalous channels are present in the MHA's Query, Key, and Value layers, where outliers vary significantly across different tokens. This

pattern is consistent in the FFB layers. We observe that disparities in weight magnitudes are less pronounced than those in activation, thus exerting a reduced impact on outlier channels. Moreover, weights distribute structurally along rows or columns (Dettmers et al., 2023; Huang et al., 2024a), affecting the overall distribution of salience from a row-wise perspective (Fig. 9). However, the most prominent salience is predominantly driven by activation across channels (column-wise).

### G.3 HESSIAN DIAGONAL CLUSTERING

Sec. 3.2.1 demonstrates that outlier tokens in input activations result in significant values at the corresponding positions along the diagonal of the weight Hessian matrix. Additionally, due to the token sink phenomenon (Xiao et al., 2023b; Nrusimha et al., 2024), areas around significantly activated key tokens exhibit increased salience, creating clusters of salient regions along the Hessian matrix diagonal. To further elucidate this phenomenon, Fig. 10 shows the values along the diagonal of the Hessian matrix for selected weights in the $2^{nd}$ and $10^{th}$ layers of the LLaMA-7B model. Within this diagonal, certain positions display pronounced values (indicated in red), whereas others are relatively moderate. In the attention aggregation layer of the $10^{th}$ layer, the token sink phenomenon results in a pronounced convergence of significant values along the Hessian matrix diagonal, with deep red areas indicating regional clustering. These findings reinforce the influence of input activations on the diagonal of the Hessian matrix, subsequently leading to a clustering phenomenon in the salience distribution of weights across channels.

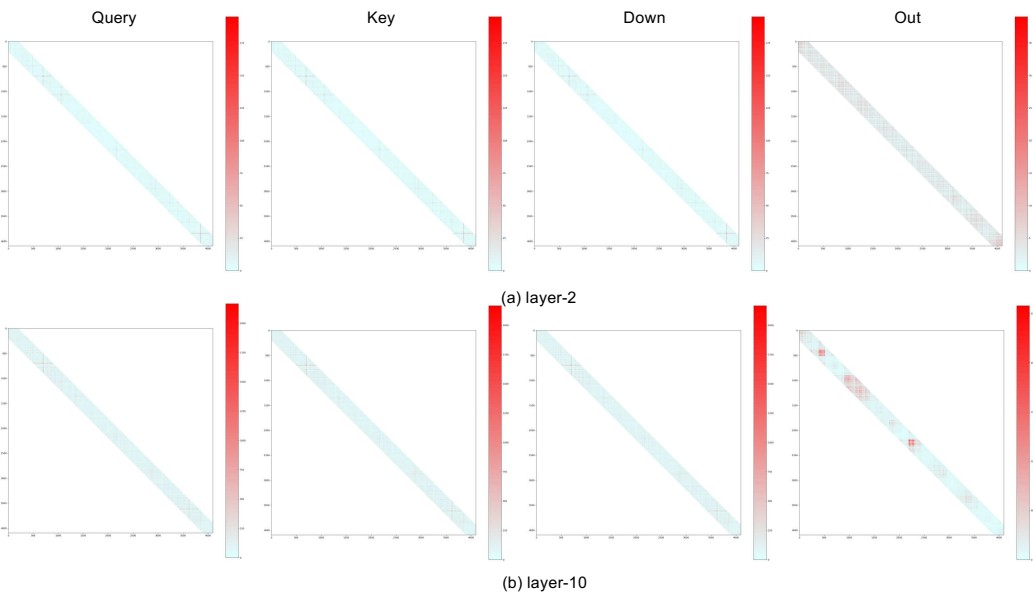

Figure 10: Hessian diagonal magnitude in attention layers of $2^{nd}$ and $10^{th}$ layers of LLaMA-7B

## H MORE COMPARISONS

In this section, we provide supplementary experiments for SliM-LLM. Tab. 8 displays the comparative results of SliM-LLM and SliM-LLM+ with other methods on the OPT series models. Tab. 9 shows the performance of SliM-LLM when quantizing the LLaMA family models on the C4 dataset, while Tab. 10 also compares the results of SliM-LLM+ on the C4 dataset. In Tab. 11, we compared the quantization results of GPTQ, AWQ, and SliM-LLM at 2-bit on the Gemma2 and Mixtral models, demonstrating the greater stability of SliM-LLM across a wider range of model structures. Additionally, in Tab. 12, we supplemented the 4-bit results of different quantization methods in the LLaMA series models, showing that SliM-LLM and SliM-LLM+ exhibit the smallest quantization errors at practical 4-bit levels. To provide a comprehensive evaluation across a broader set of benchmarks, we further compared the quantization results on MMLU and MathQA in Tab. 13.

Table 8: Quantization results of OPT Models on WikiText2 (group size is 128).

| #W PPL↓ | Method | 1.3B | 2.7B | 6.7B | 13B | 30B | 66B |
|---|---|---|---|---|---|---|---|
| 16-bit | - | 14.63 | 12.47 | 10.86 | 10.12 | 9.56 | 9.34 |
| 3-bit | RTN | 1.2e2 | 3.0e2 | 23.54 | 46.03 | 18.80 | 1.4e6 |
| | GPTQ | 16.47 | 13.69 | 11.65 | 10.35 | 9.73 | 10.96 |
| | AWQ | 16.32 | 13.58 | 11.41 | 10.68 | 9.85 | 9.60 |
| | QuIP | 16.21 | 13.79 | 11.51 | 10.50 | 9.75 | 9.59 |
| | **SliM-LLM** | **15.91** | **13.26** | **11.27** | **10.26** | **9.70** | **9.48** |
| | OmniQuant | 15.72 | 13.18 | 11.27 | 10.47 | 9.79 | 9.53 |
| | AffineQuant | 15.61 | 12.98 | 11.18 | 10.51 | 9.81 | - |
| | **SliM-LLM$^+$** | **15.58** | **12.84** | **11.18** | **10.44** | **9.67** | **9.51** |
| 2-bit | RTN | 1.3e4 | 5.7e4 | 7.8e3 | 7.6e4 | 1.3e4 | 3.6e5 |
| | GPTQ | 1.1e2 | 61.59 | 20.18 | 21.36 | 12.71 | 82.10 |
| | AWQ | 47.97 | 28.50 | 16.20 | 14.32 | 12.31 | 14.54 |
| | QuIP | 41.64 | 28.98 | 18.57 | 16.02 | 11.48 | 10.76 |
| | PB-LLM | 45.92 | 39.71 | 20.37 | 19.11 | 17.01 | 16.36 |
| | **SliM-LLM** | **30.71** | **24.08** | **14.41** | **13.68** | **11.34** | **10.94** |
| | OmniQuant | 23.95 | 18.13 | 14.43 | 12.94 | 11.39 | 30.84 |
| | **SliM-LLM$^+$** | 24.57 | **17.98** | **14.22** | **12.16** | **11.27** | 14.98 |

Table 9: Quantization results of LLaMA Family with statistic quantizer on C4 (group size is 128).

| #W PPL↓ | Method | 1-7B | 1-13B | 1-30B | 1-65B | 2-7B | 2-13B | 2-70B | 3-8B | 3-70B |
|---|---|---|---|---|---|---|---|---|---|---|
| 16-bit | - | 7.08 | 6.61 | 5.98 | 5.62 | 6.97 | 6.46 | 5.52 | 9.22 | 6.85 |
| 3-bit | APTQ | 6.24 | - | - | - | - | - | - | - | - |
| | RTN | 8.62 | 7.49 | 6.58 | 6.10 | 8.40 | 7.18 | 6.02 | 1.1e2 | 22.39 |
| | AWQ | 7.92 | 7.07 | 6.37 | 5.94 | 7.84 | 6.94 | - | 11.62 | 8.03 |
| | GPTQ | 7.85 | 7.10 | 6.47 | 6.00 | 7.89 | 7.00 | 5.85 | 13.67 | 10.52 |
| | **SliM-LLM** | **6.14** | **6.05** | **6.33** | **5.94** | **7.74** | **5.26** | **5.09** | **13.10** | 8.64 |
| 2-bit | RTN | 1.0e3 | 4.5e2 | 99.45 | 17.15 | 4.9e3 | 1.4e2 | 42.13 | 2.5e4 | 4.6e5 |
| | AWQ | 1.9e5 | 2.3e5 | 2.4e5 | 7.5e4 | 1.7e5 | 9.4e4 | - | 2.1e6 | 1.4e6 |
| | GPTQ | 34.63 | 15.29 | 11.93 | 11.99 | 33.70 | 20.97 | NAN | 4.1e4 | 21.82 |
| | QuIP | 33.74 | 21.94 | 10.95 | 13.99 | 31.94 | 16.16 | 8.17 | 1.3e2 | 22.24 |
| | PB-LLM | 49.73 | 26.93 | 17.93 | 11.85 | 29.84 | 19.82 | 8.95 | 79.21 | 33.91 |
| | **SliM-LLM** | **32.91** | **13.85** | **11.27** | **10.95** | **16.00** | **9.41** | **7.01** | 1.1e2 | **15.92** |

Table 10: Quantization results of LLaMA-1 and LLaMA-2 models with learnable quantizer on C4.

| #W PPL↓ | Method | 1-7B | 1-13B | 1-30B | 1-65B | 2-7B | 2-13B | 2-70B |
|---|---|---|---|---|---|---|---|---|
| 16-bit | - | 7.08 | 6.61 | 5.98 | 5.62 | 6.97 | 6.46 | 5.52 |
| 3-bit | OmniQuant | 7.75 | 7.05 | 6.37 | 5.93 | 7.75 | 6.98 | 5.85 |
| | AffineQuant | 7.75 | 7.04 | 6.40 | - | 7.83 | 6.99 | - |
| | **SliM-LLM$^+$** | **7.75** | **6.91** | **6.36** | 5.96 | **7.71** | **6.90** | **5.85** |
| 2-bit | OmniQuant | 12.97 | 10.36 | 9.36 | 8.00 | 15.02 | 11.05 | 8.52 |
| | AffineQuant | 14.92 | 12.64 | 9.66 | - | 16.02 | 10.98 | - |
| | **SliM-LLM$^+$** | 14.99 | **10.22** | **9.33** | **7.52** | 18.18 | **10.24** | **8.40** |

Table 11: PPL Comparison on Gemma2 and Mixtral.

| Model/Evaluation | Method | PPL (wikitext2) |
|---|---|---|
| Gemma2-9B | GPTQ 2-bit | 186.77 |
| | AWQ 2-bit | 217.83 |
| | **SliM-LLM 2bit** | **26.30** |
| Mixtral 8x7B | GPTQ 2-bit | 16.38 |
| | AWQ 2-bit | 3.2e5 |
| | **SliM-LLM 2bit** | **7.44** |

Table 12: The PPL results of our proposed method and other methods under 4bit quantization.

| Method | LLaMA-7B | LLaMA-13B | LLaMA2-7B | LLaMA2-13B | LLaMA3-8B |
|---|---|---|---|---|---|
| FP16 | 5.68 | 5.09 | 5.47 | 4.88 | 5.75 |
| AWQ | 5.81 | 5.30 | 5.62 | 4.97 | 6.63 |
| GPTQ | 5.85 | 5.20 | 5.61 | 4.98 | 6.50 |
| **SliM-LLM** | **5.83** | **5.16** | **5.59** | **4.95** | **6.42** |
| Omniquant | 5.77 | - | 5.58 | - | - |
| **SliM-LLM$^+$** | **5.75** | - | **5.57** | - | - |

Table 13: The results(%) on MMLU and MathQA for multiple quantized LLaMA models.

| Model | Method | Humanities | Social Sciences | STEM | Other | MMLU | MathQA |
|---|---|---|---|---|---|---|---|
| LLaMA-7B | GPTQ 2-bit | 24.87 | 21.84 | 21.79 | 24.01 | 23.32 | 21.11 |
| | AWQ 2-bit | 24.21 | 21.71 | 21.25 | 23.98 | 22.95 | 22.21 |
| | **SliM-LLM 2bit** | **24.94** | **23.60** | **23.40** | **25.50** | **25.10** | **23.74** |
| LLaMA-13B | GPTQ 2-bit | 24.23 | 23.20 | 22.99 | 24.78 | 23.85 | 21.68 |
| | AWQ 2-bit | 24.17 | 31.07 | 28.61 | 25.14 | 26.89 | 21.98 |
| | **SliM-LLM 2bit** | **25.12** | **31.74** | **29.19** | **26.17** | **27.05** | **23.17** |
| LLaMA2-7B | GPTQ 2-bit | 25.02 | 22.13 | 22.61 | 23.17 | 23.44 | 21.07 |
| | AWQ 2-bit | 25.12 | 22.79 | 24.26 | 24.01 | 24.51 | 19.06 |
| | **SliM-LLM 2bit** | **26.60** | **23.23** | **25.70** | **25.70** | **25.81** | **22.55** |
| LLaMA2-13B | GPTQ 2-bit | 23.91 | 27.17 | 26.10 | 25.78 | 25.53 | 20.87 |
| | AWQ 2-bit | 24.17 | 31.07 | 28.61 | 25.14 | 26.89 | 19.53 |
| | **SliM-LLM 2bit** | **26.27** | **32.20** | **29.98** | **26.46** | **27.34** | **23.48** |

## I  REAL DIALOG EXAMPLES

In this section, we show some dialogue examples of LLaMA-2-13B and Vicuna-13B with SliM-LLM-2bit and GPTQ-2bit in Fig. 11.

### LLaMA-2-13B

| | |
|---|---|
| Prompt: | The eight planets in Solar System are Mercury, ____ |
| **GPTQ-2bit**: | , 10.2. The eight pill o free games of p. 62.2. The way we do this. |
| **Slim-LLM-2bit**: | Venus, Earth, Mars, Jupiter, Saturn, Uranus and Neptune. |
| Prompt: | The tallest and most famous tower in France is ____ |
| **GPTQ-2bit**: | is 350 and most recently. |
| **Slim-LLM-2bit**: | 320 meters high. |
| Prompt: | The 2012 Summer Olympic was held in ___ |
| **GPTQ-2bit**: | The 2012 Summer Olympic was held in 1016 (a.185) and the 775,000 km from each. |
| **Slim-LLM-2bit**: | 2012 in London, the United Kingdom. |

### Vicuna-13B

| | |
|---|---|
| Prompt: | What are the capitals of Brazil, Mexico and France? |
| **GPTQ-2bit**: | What is the RER score? |
| **Slim-LLM-2bit**: | The capital of Brazil is Brasília, the capital of Mexico is Mexico City, and the capital of France is Paris. |
| Prompt: | What are the ancient civilizations in Europe? |
| **GPTQ-2bit**: | I am trying to find a book to nook over the life of to 300,000 or so. The ancient civilization is the one in which history books be pop and I have a be, oh, do you know the one the Mayan, the one of the total for Dake. |
| **Slim-LLM-2bit**: | The ancient civilizations in Europe include the Roman Empire, the Byzantine Empire, and the Ottoman Empire. |
| Prompt: | What are the logical circuits? |
| **GPTQ-2bit**: | What is the order of the logical cumulated? What is the best strategy to do in a single? What is the most popular to be eaten? What are the most sought? |
| **Slim-LLM-2bit**: | The logical circuits are the basic components for building the logical functions, which are required for digital computing. These circuits use logic structures and devices that allow the use of Boolean and multiple logic to implement the functional requirements as defined by a problem solving method of computer. In digital computing, circuits perform the logical tasks required for solving problems. |

Figure 11: Some examples of conversations. LLaMA-2-13B and Vicuna-13B are chosen to show the case of language supplementary and Q&A ability. And GPTQ-2bit is selected as the comparison. We color the text to show the reasonable or inappropriate responses.

