# OpenReview forum: "SliM-LLM: Salience-Driven Mixed-Precision Quantization for Large Language Models"
_ICLR.cc/2025/Conference — Submitted to ICLR 2025_

### Official Review · Reviewer_ouaC · 2024-10-20

**Soundness:** 2
**Presentation:** 2
**Contribution:** 2
**Rating:** 5
**Confidence:** 4

**Summary:**

This paper introduces a group-wise mixed-precision quantization method, to solve the bottlenecks of mixed-precision LLMs.
The main contribution covers two novel strategy. One is Salience-Determined Bit Allocation, named SBA. Before this, author implement many observations on Hessian values and weight salience. And determines the optimal bit-width allocation for different groups by minimizing the distance of information entropy with the original weight output. The other is Salience-Weighted Quantizer Calibration, which is named SQC, to enhance the information of significant weights within the group by amplifying the quantizer awareness of salient weight. By calibration parameter τ, optimize the quantizer, "liberating the perception interval during quantization".  Faced with problems of outlier in group, this method find a compromise, and expand the solution space of scale and zero point. The important point is scale sharing in one group.
The SBA and SQC amentioned is supported in most quantizer, including static-based and gradient-based quantizers. By Experiment result, SliM-LLM can surpass most methods in LLAMA and OPT models.

**Strengths:**

1. By observation, introduce a method to find salient weight in certain channels. By KL divergence in output activation matrix, effectively minimize the recontruction error in quantization.
2. Utilize 3 - σ rule to select salience part. By introducing τ, reduce quantization error.
3. The SQC share the parameter in group. without distinguishing weight in one group, it simplifies the memory and infer in hardware

**Weaknesses:**

1. The SBA Utilize KL divergence of block outputs as a precision allocating metric, depend on a certain calibration dataset. Generalization on datasets is not mentioned, so the method maybe effective in the same dataset.
2. The SQC introduce a novel quantizer by introducing τ to reduce reconstruction during quantization. The 3 - σ rule and 1% ratio is based on a common observation, not a heuristic method. this theoretical basis is not very strong. Dynamic ratio by layer maybe better.
3. This paper main solve the bottleneck in mixed-precision, and offers a hardware-friendly quantization. But from method and experiments section, hardware-friendly is not introduced. Including memory saving, time consuming, and superior to other method.
4.  Group-wise mixed-precision focus on different bits during same channel allocation in infer process. It is not mentioned in this paper.

**Questions:**

1. How to keep generalization on datasets, due to your observation or KL divergence, rely on calibration datasets. If I choose mmlu or humaneval, the observation is same?
2. The SQC introduce a novel method to reconstruct quantization.  what is the difference between salient weight and un-salient weight in your Formula (5), or when optimize τ, the weight or reconstruct error will be different?
3. Hardware-friendly mentioned not introduced in details. How does it save memory and time comsumption in infer process? And its advantages in this aspect, compared with PB-LLM, LLM-MQ?

---

> ### Author Response · Authors · 2024-11-25
>
> > W1: The SBA Utilize KL divergence of block outputs as a precision allocating metric, depend on a certain calibration dataset. Generalization on datasets is not mentioned, so the method maybe effective in the same dataset.
>
> Dear Reviewer ouaC,
>
> We present extensive cross-domain experimental results demonstrating that the choice of calibration dataset has minimal impact on the overall performance of LLM quantization and compression. Therefore, we follow prior high-quality quantization studies, such as GPTQ [1], by selecting the semantically rich and diverse Wikitext2 dataset for calibration, minimizing potential bias introduced during calibration.
>
> In addition to comparing the PPL performance of the quantized model on Wikitext2, we comprehensively evaluated its performance across eight zero-shot benchmarks. As shown in Table 3, our method outperforms GPTQ and OmniQuant in quantizing LLaMA-series models on datasets such as PIQA, ARC, BoolQ, Hellaswag, and Winogrande. Our results consistently lead across these datasets, indicating that calibration dataset selection did not negatively impact performance.
>
> We also note that recent work [2] has explored the effect of calibration data on LLMs. Based on the findings in [2], selecting Wikitext2 as the calibration dataset does not significantly affect final model performance. Furthermore, this slight calibration influence diminishes as model size increases.
>
> [1] Gptq: Accurate post-training quantization for generative pre-trained transformers. arXiv preprint arXiv:2210.17323 (2022).
>
> [2] On the impact of calibration data in post-training quantization and pruning. Proceedings of the 62nd Annual Meeting of the Association for Computational Linguistics (Volume 1: Long Papers). 2024.
>
> > W2: The SQC introduce a novel quantizer by introducing τ to reduce reconstruction during quantization. The 3 - σ rule and 1% ratio is based on a common observation, not a heuristic method. this theoretical basis is not very strong. Dynamic ratio by layer maybe better.
>
> We acknowledge that dynamically setting the salience ratio could potentially provide a more optimal search space. However, this approach comes with significant computational overhead. Using an empirical method such as the 3-σ rule allows for a more efficient quantization process.
>
> We greatly appreciate your valuable suggestions. While a dynamic ratio setting could indeed increase the search space and improve the likelihood of identifying a better-performing ratio, it would also incur greater search costs and computational burden. This would extend the time required to quantize FP16 mixed-precision weights into low-bit integers, potentially undermining the efficiency advantage of PTQ in terms of quantization overhead.
>
> In addition to considering the cost of dynamic ratio settings, we should emphasize that our current ratio selection method already achieves strong performance. As shown in the ablation studies in Figure 5 of our paper, the chosen ratio effectively reduces quantization error and minimizes the information loss of salient weights. This demonstrates that our current approach is both efficient and effective.
>
> That said, building on the current setup, we will continue to explore improved ratio configurations in future work.

---

> > ### Author Response · Authors · 2024-11-25
> >
> > > W3 & Q3: This paper main solve the bottleneck in mixed-precision, and offers a hardware-friendly quantization. But from method and experiments section, hardware-friendly is not introduced. Including memory saving, time consuming, and superior to other method.
> >
> > Hardware-friendly mentioned not introduced in details. How does it save memory and time comsumption in infer process? And its advantages in this aspect, compared with PB-LLM, LLM-MQ?
> >
> > We would like to clarify that in Table 5 of our manuscript, we provide a detailed description of SliM-LLM’s memory, latency, and accuracy during inference. Our speed and memory consumption are nearly identical to those of non-mixed-precision quantization methods, while achieving better performance. As mentioned in Section 1, existing unstructured mixed-precision quantization methods place a heavy burden on hardware. This is because such methods introduce additional storage requirements, such as bitmap or code index storage, and require extra computations to decode these bitmaps or indexes, which reduces hardware friendliness.
> >
> > In contrast, we achieve mixed-precision quantization through structured grouping, utilizing the observed grouping properties of the salience distribution. This allows us to efficiently split groups and search for optimal quantization configurations for each group, reducing the hardware pressure caused by previous methods, while further improving the performance of ultra-low-bit mixed-precision quantization.
> >
> > Additionally, in Appendix A.2, we explain the computational optimizations on CUDA. For instance, we pack the quantized integers from different groups together, unpack them during computation, and fetch them with their corresponding scales to perform parallel scheduling. This approach optimizes computations and makes it more hardware-friendly.
> >
> > Compared to PB-LLM and LLM-MQ, a significant difference is that these methods are unstructured and require an additional 1-bit mask to define precision positions, whereas our method is structured. In our approach, parameter settings within each group are shared and organized, avoiding the sparse data indexing required by unstructured methods. Furthermore, the methods they propose primarily involve fake quantization, which is not hardware-friendly and difficult to implement in hardware. Our structured grouping method enables rapid hardware implementation.
> >
> >
> > > W4: Group-wise mixed-precision focus on different bits during same channel allocation in infer process. It is not mentioned in this paper.
> >
> > We would like to clarify that in a weight matrix of size $n\times m$, when the group size is 128, the weight size of each group is $n\times 128$, with all having the same bit-width. This structured grouping and bit-width allocation is a core efficiency advantage that we emphasize. In per-channel quantization, an $n\times 128$ group matrix will have $n$ sets of quantization parameters, which is exactly the same setup as GPTQ[1], AWQ[2], and OmniQuant[3], with no additional parameter consumption compared to these methods. This approach allows for full structuring, slicing the weight matrix.
> >
> > Therefore, SliM-LLM allocates different precisions to each group at this granularity, making it highly hardware-friendly for both weight storage and actual inference. Regarding this, you can also refer to the appendix, which defines and explains the data packaging and computation. After quantizing FP16 data to ultra-low-bit integers, we pack them into normal 32-bit integers to compress space, while aligning and storing the corresponding quantization parameters. During parallel computation, the data can be unpacked using the quantization parameters and then dequantized, which benefits model inference.
> >
> > [1]Frantar, Elias, et al. "Gptq: Accurate post-training quantization for generative pre-trained transformers." arXiv preprint arXiv:2210.17323 (2022).
> >
> > [2]Lin, Ji, et al. "AWQ: Activation-aware Weight Quantization for On-Device LLM Compression and Acceleration." Proceedings of Machine Learning and Systems 6 (2024): 87-100.
> >
> > [3]Shao, Wenqi, et al. "Omniquant: Omnidirectionally calibrated quantization for large language models." arXiv preprint arXiv:2308.13137 (2023).

---

> > > ### Author Response · Authors · 2024-11-25
> > >
> > > > Q1: How to keep generalization on datasets, due to your observation or KL divergence, rely on calibration datasets. If I choose mmlu or humaneval, the observation is same?
> > >
> > > To address the issue you mentioned, we conducted additional tests on several LLaMA models calibrated using Wikitext2, evaluating their performance with different ultra-low-bit quantization methods on MMLU and MathQA. The experimental results are shown in Table 1 below.
> > >
> > > Table 1 The results on MMLU and MathQA for multiple quantized LLaMA models
> > >
> > > | **Model/Evaluation** | **Method**     | **humanities** | **social_sciences** | **stem**  | **other** | **mmlu**  | **Mathqa** |
> > > | -------------------- | -------------- | -------------- | ------------------- | --------- | --------- | --------- | ---------- |
> > > | Llama-7b             | GPTQ 2-bit     | 24.87          | 21.84               | 21.79     | 24.01     | 23.32     | 21.11      |
> > > |                      | AWQ 2-bit      | 24.21          | 21.71               | 21.25     | 23.98     | 22.95     | 22.21      |
> > > |                      | SliM-LLM 2-bit | **24.94**      | **23.60**           | **23.40** | **25.50** | **25.10** | **23.74**  |
> > > | Llama-13b            | GPTQ 2-bit     | 24.23          | 23.20               | 22.99     | 24.78     | 23.85     | 21.68      |
> > > |                      | AWQ 2-bit      | 24.17          | 31.07               | 28.61     | 25.14     | 26.89     | 21.98      |
> > > |                      | SliM-LLM 2-bit | **25.12**      | **31.74**           | **29.19** | **26.17** | **27.05** | **23.71**  |
> > > | Llama-2-7b           | GPTQ 2-bit     | 25.02          | 22.13               | 22.61     | 23.17     | 23.44     | 21.07      |
> > > |                      | AWQ 2-bit      | 25.12          | 22.79               | 24.26     | 24.01     | 24.51     | 19.06      |
> > > |                      | SliM-LLM 2-bit | **26.60**      | **23.23**           | **25.70** | **25.70** | **25.81** | **22.55**  |
> > > | Llama-2-13b          | GPTQ 2-bit     | 23.91          | 27.17               | 26.10     | 25.78     | 25.53     | 20.87      |
> > > |                      | AWQ 2-bit      | 24.17          | 31.07               | 28.61     | 25.14     | 26.89     | 19.53      |
> > > |                      | SliM-LLM 2-bit | **26.27**      | **32.20**           | **29.98** | **26.46** | **27.34** | **23.48**  |
> > >
> > > As observed, our method outperforms GPTQ[1], AWQ[2], and other state-of-the-art 4-bit quantization methods on most metrics, including STEM, Other, and overall performance on MMLU and MathQA. This demonstrates the generalization capability of our approach, achieving competitive results even under a 2-bit ultra-low-bit setting, thereby supporting the same conclusions.
> > >
> > > [1]Frantar, Elias, et al. "Gptq: Accurate post-training quantization for generative pre-trained transformers." arXiv preprint arXiv:2210.17323 (2022).
> > >
> > > [2]Lin, Ji, et al. "AWQ: Activation-aware Weight Quantization for On-Device LLM Compression and Acceleration." Proceedings of Machine Learning and Systems 6 (2024): 87-100.
> > >
> > > > Q2: The SQC introduce a novel method to reconstruct quantization. what is the difference between salient weight and un-salient weight in your Formula (5), or when optimize τ, the weight or reconstruct error will be different?
> > >
> > > The distinction between salient and non-salient weights is primarily based on the $3\sigma$ principle, which assesses the relationship between the salience value and the $3\sigma$ boundary, using the salience values of weights within a group to classify them as salient or non-salient. Regarding $\tau$ in Equation 5, since our quantization error objective does not treat all weights as equal when calculating the average error, SQC separates the salient and non-salient weights. This allows them to contribute equally to the objective. Therefore, $τ$ is used to regulate the scaling factor in the quantizer, ensuring that the objective finds the minimum error across both salient and non-salient weights.
> > >
> > > > Q3: Hardware-friendly mentioned not introduced in details. How does it save memory and time comsumption in infer process? And its advantages in this aspect, compared with PB-LLM, LLM-MQ?
> > >
> > > Please check the reply for Weakness3.

---

> > > > ### Comment · Reviewer_ouaC · 2024-11-26
> > > >
> > > > Thanks for authors' response, which has addressed some of my concerns. I still have some confusion.
> > > > 1. To W3 & Q3, authors mentioned Tab. 5 deployment results which show your method advantages. But Tab. 5 is not cited in the whole paper. On the other hand, SLIM-LLM is superior to GPTQ in terms of ppl, but in memory and latency, the speed maybe 20% slower  than ordinary GPTQ, and deployment results with PB-LLM or LLM-MQ in the same ratio not mentioned. So this table maybe incomplete?
> > > > 2. To Q1, authors relative experiments are supplemented, MMLU and MathQA are multiple-choice questions. Most quantizers can not show obvious gap in these questions, especially in groupwise quantizers. And another question is: during KL Divergence, how to process some corner case, for example, in 2 2048x4096 tensors, if some outliers in tensors, the original kl value maybe show the similarity of outlier distribution, how about other small values? And if the logarithmic operation overflows, how to avoid occurrence of the above situation?

---

> > > > > ### Author Response · Authors · 2024-11-26
> > > > > **Reply to Reviewer**
> > > > >
> > > > > Dear reviewer ouaC,
> > > > >
> > > > > Thank you for your kindly reminder. Tab.5 should be referenced in Section 4.3, and we will correct this typo in the revised version.
> > > > >
> > > > > - Yes, as you have observed, our speed is slightly slower compared to GPTQ, but our tested speed of 61.2 Token/s is notable for **its transparency and practicality** due to our significant accuracy improvements. We attribute GPTQ's speed advantage to our comparison with the most advanced GPTQ CUDA kernel in the current AutoGPTQ toolchain. Initially, GPTQ's speed at its release did not match present data, **as its hardware acceleration has undergone multiple iterations over the past two years**. One of our **future optimization goals includes enhancing the group-wise mixed-precision CUDA kernel, which may inspire hardware developers to improve more efficient group-wise mixed-precision inference kernels in the future.**
> > > > > - The various unstructured mixed-precision quantization methods we mentioned, such as PB-LLM and LLM-MQ, face challenges in achieving real inference on GPUs due to their unstructured bit-width allocation methods. **Also, the lack of real inference deployment in previous articles and code repositories hinders our comparison of speed.** Hence, our Tab.5 is comprehensive. **On the other hand, our SliM-LLM not only deploys efficiently on GPUs but also outperforms existing unstructured mixed-precision PTQ methods in PPL across multiple evaluation metrics.**
> > > > > - We chose WikiText2 as our SBA searching calibration because it represents a broader range of general language content with minimal biases, and it has been commonly used for calibration in previous works. In Tab.3 of our paper, we also presented the results of six zero-shot benchmarks, where SliM-LLM consistently maintains a fair and stable lead in accuracy. In response to Q1, we followed the suggestion to include a more comprehensive view by showcasing the MMLU and MathQA ( math as domain-specific benchmarks), aiming to further validate the robustness of SliM-LLM across diverse datasets.
> > > > > - Great question! We have indeed observed the occurrence of outlier values in activations, as you mentioned. These outliers can lead to extreme distributions in KL results, causing very small values at other data points. This outcome results in the SBA method tending to maintain these outlier distribution with searching the optimal bit allocation. Interestingly, **this ultimately contributes positively to LLM performance, as outliers in LLM have been proven to play a significant role in model performance**. Regarding the issue of logarithmic operation overflows, we have noted that while the **OPT series models use standard ReLU activation, modern Llama and other new model architectures employ smoother activation functions and normalization mechanisms**. Consequently, among various LLM structures, **OPT activations exhibit the most prominent outliers.** However, our experiments have shown that the KL Divergence used by SBA does not lead to logarithmic operation overflows in OPT models, with a low likelihood of such occurrences in other models.
> > > > >
> > > > > We hope that our response further clarifies your question. Thank you for your time.

---

> > > > > > ### Comment · Reviewer_ouaC · 2024-12-02
> > > > > >
> > > > > > Thanks for the authors' response, which has addressed most of my concerns. But in experiments of perfomance and generalization, including some aspects of mathematical reasoning, I don't fully agree with this view. I maintain my original positive score.

---

### Official Review · Reviewer_jmUu · 2024-11-03

**Soundness:** 3
**Presentation:** 3
**Contribution:** 2
**Rating:** 5
**Confidence:** 4

**Summary:**

The paper introduces SliM-LLM, a framework for low-bit quantization of Large Language Models (LLMs). The key innovation is a mixed-precision approach that uses two main components:
1. Salience-Determined Bit Allocation: Hessian-guided assigns bit widths to groups of weights based on their importance
2. Salience-Weighted Quantizer Calibration: Optimizes quantization while preserving important weights

The method achieves significant compression (2-bit quantization) while maintaining model performance and ensuring hardware-friendly deployment. The framework is validated across multiple LLM families and tasks, showing practical benefits for deploying LLMs in resource-constrained environments.

**Strengths:**

1. Introduces an innovative two-component framework combining global group-level and local element-level salience
2. The structured approach is hardware-friendly while maintaining good performance. It achieves actual GPU acceleration with reasonable inference speed.
3. Comprehensive experiments across multiple LLM families, showing improvements over existing methods.

**Weaknesses:**

1. The equations in this paper have significant problems:
- In Equation 2, the $\alpha$ is not used. The correct equation may be $\hat{w}_b=\alpha \text{sign}(w_f)$.
- In Equation 3, the equation does not hold, please confirm if it is an approximation.
- In Equation 4, using KL divergence for outputs is mathematically incorrect as they aren't probability distributions. Variables g1,...,gn are undefined. The definition of $w_f^{sba}$ needs proper notation (The superscript of $w^{b_0}$ cannot denote for bitwidth selection).
- In Equation 5, the braces are not use appropriately.

﻿2. The main problem of this paper is that the algorithm is poorly explained:
﻿- Critical algorithm details are hidden in appendix. Implementation details are unclear.
﻿- Salience-weighted quantizer calibration is not well-explained, which is hard to understand. In Section 3.2.2, the search space for s and z is undefined. And the role of τ in expanding solution space isn't clearly justified. The authour should clarify how τ expands the solution space and why this is beneficial.

﻿3. Limited comparison with other extremely low-bit quantization methods. For instance, "Ma S, Wang H, Ma L, et al. The Era of 1-Bit LLMs: All Large Language Models Are in 1.58 Bits [J]. arXiv preprint arXiv:2402.17764, 2024." This method is popular and user-friendly, as it is supported by Hugging Face.

**Questions:**

1. Is PTQ practical for ultra-low bit quantization in the case of significant dropout?
2. Is there any details about how exactly does τ expand the solution space of s and z?
3. How do you justify using KL divergence as objective function?

---

> ### Author Response · Authors · 2024-11-25
>
> > W1: In Equation 2, the $\alpha$ is not used. The correct equation may be $\hat w_b=\alpha \operatorname{sign}(w_f)$.
>
> > In Equation 3, the equation does not hold, please confirm if it is an approximation.
>
> > In Equation 4, using KL divergence for outputs is mathematically incorrect as they aren't probability distributions. Variables g1,...,gn are undefined. The definition of $w^{sba}_f$ needs proper notation (The superscript of $w^{b_0}$ cannot denote for bitwidth selection).
>
> > In Equation 5, the braces are not use appropriately.
>
> Dear Reviewer jmUu,
>
> We would like to clarify the point regarding Equation 2. As stated in line 185, we have defined $\hat w_b$ as the binarized results, while the $\alpha\operatorname{sign}(w_f)$ you mentioned refers to the float matrix weights after dequantization. We will include the float matrix weights in the final description as you suggested.
>
> Regarding Equation 3, we considered the optimal scenario where the network training converges, and thus we equated the second-order error of the weight output to the right-hand side of Equation 3. This description follows the definitions in [1], where Equation 1 and Equation 2 are defined using $=$. We fully agree with your more rigorous definition, and in the revised version, we will update this to $\approx$ instead of $=$.
>
> In the actual computation, we first apply softmax to the weight output features and then calculate the KL divergence. We will follow your suggestion and present this calculation process in the revised Equation 4. Additionally, the definitions of $g_1, \dots, g_n$ as the structured groups in the weight matrix will be clearly specified in the revised version, along with the modification of the expression for $w^{b_0}$.
>
> Thank you for your valuable suggestions. We will also update the notation in Equation 5 by replacing $\lbrace\rbrace$ with $()$.
>
> [1] QuIP#: Even Better LLM Quantization with Hadamard Incoherence and Lattice Codebooks. ICML 2024.
>
> > W2 & Q2:  The main problem of this paper is that the algorithm is poorly explained: ﻿- Critical algorithm details are hidden in appendix. Implementation details are unclear. ﻿- Salience-weighted quantizer calibration is not well-explained, which is hard to understand. In Section 3.2.2, the search space for $s$ and $z$ is undefined. And the role of $τ$ in expanding solution space isn't clearly justified. The authour should clarify how $τ$ expands the solution space and why this is beneficial.
>
> Thank you for your feedback. We have provided a detailed explanation of the use of $\tau$ in Section 3.3.2, lines 343-346. As shown in Equation 1, the general calculation of $s$ and $z$ is derived from the statistical values of all weight elements. However, this is often not the optimal quantization approach. To improve the representation of salient weights, SQC introduces Equation 5 to resolve for $s$ and $z$. Specifically, as defined in line 345, we expand the search range for $\tau$ to $[1 - \lambda, 1 + \lambda]$, dividing it into $2n$ evenly spaced points. Therefore, $\tau$ will fluctuate within this range to satisfy the objective in Equation 5. Under the influence of $\tau$, the search space for ss will be adjusted to $[(1 - \lambda) s, (1 + \lambda) s]$, with $2n$ search elements.

---

> ### Author Response · Authors · 2024-11-25
>
> > W3: Limited comparison with other extremely low-bit quantization methods. For instance, "Ma S, Wang H, Ma L, et al. The Era of 1-Bit LLMs: All Large Language Models Are in 1.58 Bits [J]. arXiv preprint arXiv:2402.17764, 2024." This method is popular and cuser-friendly, as it is supported by Hugging Face.
>
> The paper *The Era of 1-Bit LLMs* is indeed a valuable contribution to the field of low-bit quantization for LLMs, providing the community with a method for ultra-low-bit quantization and making it openly available on platforms such as Huggingface. However, unlike our work, this paper employs a QAT approach that involves training from scratch, which typically requires significantly more time and GPU resources to quantize the weights.
>
> In contrast, our PTQ method only requires 50 minutes on a single GPU, offering a different and orthogonal approach to exploring ultra-low-bit quantization for LLMs. The two methods are not mutually exclusive. Our work focuses specifically on PTQ-based ultra-low-bit mixed-precision quantization for LLMs, as detailed in Section 1 of our paper.
> Thank you for your suggestion. We will include a citation and discussion of this paper in our revised manuscript, highlighting the distinctions and connections between our method and QAT.
>
> > Q1: Is PTQ practical for ultra-low bit quantization in the case of significant dropout?
>
> We believe that PTQ research in the field of ultra-low-bit quantization is highly valuable, especially in resource-constrained environments. While existing PTQ methods have achieved significant success at conventional bit widths (>3-bit), ultra-low-bit quantization still requires exploration of more effective solutions.
>
> As shown in Table 3 of our paper, our method demonstrates notable improvements over others at ultra-low-bit settings, achieving around 2-point gains on various benchmarks such as PIQA and ARC compared to GPTQ[1] and AWQ[2]. Additionally, as stated in the paper, our structured quantization approach is hardware-friendly, further supporting the effectiveness of our work in advancing PTQ quantization performance at ultra-low bit widths.
>
> Although QAT methods, which involve training on more comprehensive datasets, can achieve better performance, their training time is typically measured in days. In contrast, PTQ offers high efficiency, low resource usage, and the ability to run end-to-end on resource-constrained devices. Based on our findings, we believe PTQ still has significant potential for further exploration in compression.
>
> Through this work, we aim to provide more insights and practical experience in PTQ quantization to the community, contributing to the advancement of this promising technology.
>
> [1]Frantar, Elias, et al. "Gptq: Accurate post-training quantization for generative pre-trained transformers." arXiv preprint arXiv:2210.17323 (2022).
>
> [2]Lin, Ji, et al. "AWQ: Activation-aware Weight Quantization for On-Device LLM Compression and Acceleration." Proceedings of Machine Learning and Systems 6 (2024): 87-100.
>
> > Q2: Is there any details about how exactly does $τ$ expand the solution space of $s$ and $z$?
>
> Please check the reply for Weakness2.
>
> > Q3: How do you justify using KL divergence as objective function?
>
> Before applying the practical KL divergence, we first transform the weight outputs into a probability distribution using softmax and then perform the computation. We believe that KL divergence, based on the concept of information theory, provides a measure of the information content between two distributions. This makes it particularly useful in certain cases, offering a deeper understanding of distribution differences, especially in the construction of feature maps in the deeper hidden layers of the model. We compared it with the commonly used second-order loss in the table below and found that KL divergence, as an objective for assigning bit-widths, is more robust and achieves better performance:
>
> Table 1 Comparison between KL divergence and MSE
> | Method           | #W   | OPT-1.3B  | OPT-2.7B  | OPT-6.7B  | OPT-13B   | LLaMA-7B  | LLaMA2-7B |
> | ------------- | ---- | --------- | --------- | --------- | --------- | --------- | --------- |
> | MSE           | 2bit | 32.50     | 27.58     | 15.14     | 13.28     | 21.94     | 16.86     |
> | KL Divergence | 2bit | **30.71** | **13.26** | **11.27** | **10.12** | **14.58** | **16.01** |

---

> > ### Comment · Reviewer_jmUu · 2024-11-29
> > **Raise the score for revised version**
> >
> > After reviewing the revised version, I found that most of the issues with the equations were addressed. Additionally, the new results answered my questions. I have increased my rating from 3 to 5.

---

### Official Review · Reviewer_rUsM · 2024-11-03

**Soundness:** 3
**Presentation:** 3
**Contribution:** 1
**Rating:** 5
**Confidence:** 4

**Summary:**

The paper proposes SliM-LLM, a quantization approach for LLMs that employs a salience-driven mixed-precision scheme. The key idea is to allocate bit-widths dynamically, giving more precision on model weights that are "salient". SliM-LLM introduces two main components: Salience-Determined Bit Allocation (SBA), which assigns bit-widths to groups within each layer based on their group-level salience to reduce reconstruction error, and Salience-Weighted Quantizer Calibration (SQC), which fine-tunes the quantization parameters by emphasizing highly influential weights. Experiments demonstrate SliM-LLM's effectiveness in various LLMs, while with noticeable slowdown (Token/s) on GPUs running 7B models.

**Strengths:**

The experiments are thorough, spanning multiple models and bit configurations.

The paper explains both the SBA and SQC components of the approach well, though some additional clarity in experimental detail would help.

**Weaknesses:**

Inference slow down: the paper suffers from a significant problem that the inference speed of the Llama2-7B quantized by SLIM-LLM is even slower than the fp16 baseline (Table 5). Users may as well not do any SLIM-LLM quantization: worse PPL and worse speed compared to fp16 baseline.

**Questions:**

The 2-bit PPL is still much higher compared to fp16, it's not practical for practical use yet. 4-bit weight is still the most widely used deployment case. Can the authors provide 4-bit PPL comparison with RTN, AWQ and GPTQ?

---

> ### Author Response · Authors · 2024-11-25
>
> > W1: Inference slow down: the paper suffers from a significant problem that the inference speed of the Llama2-7B quantized by SLIM-LLM is even slower than the fp16 baseline (Table 5). Users may as well not do any SLIM-LLM quantization: worse PPL and worse speed compared to fp16 baseline.
>
> Dear Reviewer rUsM,
>
> We would like to clarify that our method primarily focuses on significantly reducing memory usage during LLM inference at ultra-low bit-widths while maintaining inference performance. This enables the deployment of large-scale LLMs in resource-constrained edge scenarios. As shown in Table 5, our mixed-precision quantization method reduces memory usage to 16% of FP16 in the 2-bit quantization experiment for LLaMA-13B. Additionally, compared to other methods, our approach demonstrates significant improvements in PPL performance at ultra-low bit-widths. For instance, in Table 5, our method achieves nearly a 12 point improvement in PPL over GPTQ for 2-bit quantization of LLaMA-13B.
>
> Regarding the inference speed issue you mentioned, weight-only quantization involves complex dequantization operations, which indeed introduce some additional computation and slightly reduce inference speed. However, after thoroughly investigating existing quantization frameworks such as AutoGPTQ, we observed that inference speed often decreases under 2-bit quantization due to the finer-grained weight packing required. We believe that hardware support for ultra-low bit-width quantization remains a promising area of research, and further optimizing the hardware compatibility of ultra-low-bit quantization will be an important focus of our future work.
>
> > Q1: The 2-bit PPL is still much higher compared to fp16, it's not practical for practical use yet. 4-bit weight is still the most widely used deployment case. Can the authors provide 4-bit PPL comparison with RTN, AWQ and GPTQ?
>
> In comparison, our method does result in relatively higher PPL at 2-bit quantization compared to FP16. However, it represents a significant improvement over existing methods in optimizing 2-bit quantization for LLMs. As shown in Table 5, our approach achieves better performance under ultra-low-bit quantization than other frameworks, advancing research in this area and demonstrating the great potential and value of ultra-low-bit quantization.
>
> Regarding your suggestion about comparative experiments with 4-bit quantization, we sincerely appreciate your input. We would like to clarify that while SliM-LLM is primarily designed for 2-bit and 3-bit quantization, our method is also compatible with 4-bit mixed-precision quantization. We have added experiments in the revised Table 1 and Table 2 below to address this. And we also tried it on some 4-bit models, we are happy to share the results to you:
>
> Table 1 The results of our proposed method and other methods under 4bit quantization
>
> | **Method** | **LLaMA-7B** | **LLaMA-13B** | **LLaMA2-7B** | **LLaMA2-13B** | **LLaMA3-8B** |
> | ---------- | ------------ | ------------- | ------------- | -------------- | ------------- |
> | FP16       | 5.68         | 5.09          | 5.47          | 4.88           | 5.75          |
> | AWQ        | 5.81         | 5.30          | 5.62          | 4.97           | 6.63          |
> | GPTQ       | 5.85         | 5.20          | 5.61          | 4.98           | 6.50          |
> | SliM-LLM   | 5.83         | 5.16          | 5.59          | 4.95           | 6.42          |
>
>
>
> | **Method** | **LLaMA-7B** | **LLaMA2-7B** |
> | ---------- | ------------ | ------------- |
> | Omniquant  | 5.77         | 5.58          |
> | SliM-LLM+  | 5.75         | 5.57          |

---

> > ### Comment · Reviewer_rUsM · 2024-11-25
> >
> > I would keep the original score. I'm not convinced by worsening the PPL by **more than 10 points** for llama2-7B, while being **slower** (lower throughput) using the proposed 2bit method compared to FP16, despite the memory is reduced to 16%. A better solution should using a model with fewer parameters, but 4bit quantization.

---

> > > ### Author Response · Authors · 2024-11-26
> > > **Further alleviate of reviewer's questions**
> > >
> > > Dear reviewer rUsM,
> > >
> > > Thank you for your timely response, and we hope to further  alleviate your concerns. Our SliM-LLM is a plug-and-play method that significantly enhances the performance of **existing group-wise PTQ quantization methods at various bit-widths** through the configuration of mixed-precision group SBA and SQC technologies. We primarily emphasize 2-bit and 3-bit quantization in the main text because SliM-LLM excels in accuracy at ultra-low bit-widths. In reality, as we observed in the rebuttal Table, our method also demonstrates **competitive advantages at 4 bits**, further proving the practicality of SLiM-LLM.
> > >
> > > Regarding the trade-off between speed and accuracy that you mentioned, **when we integrate SliM-LLM+ into Omniquant as shown in Table 2, we achieve a PPL of 10.87 on llama2-7b and 7.59 on llama2-13b, indicating a significant improvement over SliM-LLM (GPTQ+SBA+SQC)**. During actual inference, SliM-LLM+ maintains the same speed as SliM-LLM since their inference structures are identical. We want to emphasize that the group-wise mixed-precision approach proposed by SliM-LLM can notably enhance PTQ quantization accuracy across different models and bit-widths.
> > >
> > > Regarding speed, despite being deployed in existing 2-bit PTQ works, our actual deployment speed remains practical. We have observed that some unstructured works and codebook methods can greatly impact model inference speed, sometimes hindering transparent speed testing. Our tested speed of 61.2 Token/s **is both transparent and applicable**. We fully agree with your mentioned importance of actually decoding speed, and, optimizing the group-wise mixed-precision CUDA kernel is one of our future optimization goals, which may also inspire hardware developers to create more efficient group-wise mixed-precision inference kernels.
> > >
> > > Taking into account aspects such as memory, accuracy, and speed, we are positive that the structured mixed-precision technique introduced by SliM-LLM offers valuable contributions to the field. We kindly request your reconsideration of the contribution of SliM-LLM, with hopes of improving your rating.

---

### Official Review · Reviewer_pHSj · 2024-11-04

**Soundness:** 3
**Presentation:** 3
**Contribution:** 3
**Rating:** 6
**Confidence:** 3

**Summary:**

This paper proposes salience-aware bit allocation and calibration methods for low-bit quantization of large language models (LLMs). Unlike conventional uniform bit allocation, this method allocates bit-width based on the weight salience of each group. Additionally, the quantizer calibration incorporates element-level salience, enhancing overall model quality.

**Strengths:**

•	The fine-grained bit allocation and calibration method demonstrates effectiveness, achieving improved quality compared to existing post-training quantization baselines.

•	The paper implements inference support for this new quantization algorithm.

**Weaknesses:**

•	The authors are suggested to evaluate and report the quantization cost, since the proposed approaches introduce searching overhead.

•	While the theoretical expectation for inference speedup from 16-bit to 2-bit quantization is 8x, the actual inference speed of the proposed approach can even be slower than its FP16 counterpart. This makes the proposed approach less convincing. It may be beneficial to explore state-of-the-art low-bit inference frameworks, such as T-MAC, to see if the issue can be addressed.

•	Figure 4 lacks clarity in its legends and scales, making it difficult to interpret.

**Questions:**

Since low-bit quantization is popularly used for the LLM deployment on resource-limited user devices, such as laptops and mobile phones. How would you expect the results will be on these devices.

---

> ### Author Response · Authors · 2024-11-25
>
> > W1: The authors are suggested to evaluate and report the quantization cost, since the proposed approaches introduce searching overhead.
>
> Dear Reviewer pHSj,
>
> Thank you for your suggestion. In fact, when applying SBA and SQC for PTQ mixed-precision quantization under single-GPU edge conditions, the quantization process for a 7B model takes only about 50 minutes. This is significantly more efficient compared to QAT methods, which typically require multiple GPUs and take a day or more. We will include details about our offline quantization time in the revised version to highlight the efficiency of our approach in optimizing quantization overhead.
>
> > W2: While the theoretical expectation for inference speedup from 16-bit to 2-bit quantization is 8x, the actual inference speed of the proposed approach can even be slower than its FP16 counterpart. This makes the proposed approach less convincing. It may be beneficial to explore state-of-the-art low-bit inference frameworks, such as T-MAC, to see if the issue can be addressed.
>
> Our method focuses on reducing memory usage during LLM inference at ultra-low bit widths, enabling the deployment of large-parameter LLMs in resource-constrained edge scenarios. As shown in Table 5 of the paper, our mixed-precision quantization method reduces memory usage to 16% of FP16 when performing 2-bit mixed-precision quantization on LLaMA-13B. Additionally, compared to other methods like GPTQ, our approach achieves significant improvements in perplexity (PPL) at ultra-low bit widths. For example, as presented in Table 5 of our manuscript, our 2-bit quantization of LLaMA-13B achieves approximately a 12-point reduction in PPL compared to GPTQ.
>
> In terms of runtime efficiency, weight-only quantization often involves complex dequantization operations, which may introduce some additional computation and slightly reduce inference speed. Through a thorough investigation of existing GPU-based quantization frameworks such as AutoGPTQ, we observed that at 2-bit quantization, these methods often experience speed reductions due to finer-grained weight packing. Recognizing this, we believe that hardware support for ultra-low bit widths remains a promising research direction. Further optimizing hardware friendliness for ultra-low-bit quantization will also be a focus of our future work.
>
> > W3: Figure 4 lacks clarity in its legends and scales, making it difficult to interpret.
>
> We apologize for any inconvenience caused by the insufficient explanations. We will further improve and complete the annotations for all figures in future revisions to enhance readability.
>
>
> > Q1: Since low-bit quantization is popularly used for the LLM deployment on resource-limited user devices, such as laptops and mobile phones. How would you expect the results will be on these devices.
>
> We are optimistic that our method, along with other similar ultra-low-bit-width LLM quantization and compression strategies, could be deployed on laptops and mobile phones in the future. We observe that most PTQ-based studies focus on optimizing compression on resource-constrained edge devices, such as single-GPU setups, aiming to minimize resource requirements during offline quantization and post-quantization inference while ensuring storage and inference efficiency.
> Some existing works, such as AWQ [1] and GPTQ [2], have already been successfully adapted for deployment on mobile and laptop devices. These methods leverage salience detection, scaling, and structured grouping strategies to achieve hardware friendliness, maintain inference performance, and enable low compression ratios. They are often integrated with optimized frameworks for mobile SoCs, such as ollama and bitsandbytes, which are well-suited for mobile-oriented quantization tasks.
> Given the hardware-friendly nature of our approach and the choice of group-wise quantization, we believe our method can also be seamlessly integrated into these toolchains, enabling potential deployment on mobile and laptop platforms.
>
> [1]Lin, Ji, et al. "AWQ: Activation-aware Weight Quantization for On-Device LLM Compression and Acceleration." Proceedings of Machine Learning and Systems 6 (2024): 87-100.
>
> [2]Frantar, Elias, et al. "Gptq: Accurate post-training quantization for generative pre-trained transformers." arXiv preprint arXiv:2210.17323 (2022).

---

### Official Review · Reviewer_mxH3 · 2024-11-05

**Soundness:** 2
**Presentation:** 2
**Contribution:** 2
**Rating:** 6
**Confidence:** 4

**Summary:**

The paper presents "SliM-LLM," a quantization method for large language models (LLMs), focusing on salience-driven, mixed-precision quantization to enhance memory efficiency and maintain model performance at low bit-widths. The proposed framework integrates two main steps: Salience-Determined Bit Allocation (SBA), which dynamically allocates bit-widths to weights based on group-wise importance, and Salience-Weighted Quantizer Calibration (SQC), which aims to further improve quantization quality by adjusting quantizer parameters based on local salience. Overall experiments demonstrate that SliM-LLM significantly improves performance metrics (e.g., perplexity) across various model sizes with Llama series, outperforming several state-of-the-art quantization methods in both accuracy and efficiency.

**Strengths:**

- Clear problem definition and motivation
- Good visualization
- Comprehensive results in various model size

**Weaknesses:**

- The paper lacks a transparent discussion of potential limitations, such as possible trade-offs between compression rate and inference speed, or cases where the approach may not perform as well, such as for models with different architectural peculiarities. Acknowledging these limitations would strengthen the overall rigor of the paper.
- The experimental evaluation in this paper primarily focuses on the LLaMA series, with additional results on the OPT model series presented in the appendix. However, there is little discussion or empirical validation on architectures beyond these two families, which share many similarities in their transformer structures. This limited scope raises concerns about the generalizability of SliM-LLM across more diverse architectures. The absence of empirical results on these varied architectures leaves open the question of how well the proposed salience-based quantization strategies would perform in models with different salience and outlier channel dynamics. Extending experiments to a broader set of architectures would strengthen the claim of SliM-LLM’s robustness and applicability across LLMs.
- The proposed SliM-LLM approach is designed based on empirical observations of outlier channels and salience distributions in models like LLaMA. However, recent architectures such as Gemma2 incorporate normalization layers following both attention and feed-forward (FFN) modules, which could alter the distribution and behavior of outlier channels. These normalization layers might dampen or redistribute the significance of certain activations, potentially influencing the effectiveness of salience-based quantization strategies. This variation raises questions about how well SliM-LLM’s bit allocation and salience-weighted calibration would generalize to such architectures, where outlier handling may not align with observations made in models without inter-layer normalization. A more in-depth analysis would help determine if additional adaptations are required to support architectures that use different normalization schemes, as seen in models like Gemma2.

**Questions:**

- The paper briefly mentions spatial clustering of salient weights, but how does the model respond to cases where salience does not cluster as neatly? Are there adaptive mechanisms in SBA and SQC to handle such irregularities?
- Given the additional steps introduced in SBA and SQC, how does the computational cost compare to baseline quantization methods? Specifically, are there latency trade-offs that should be considered, especially for real-time applications?
- The paper focuses on transformers for text, but given the growing interest in multi-modal LLMs, could this quantization approach generalize to models that handle other data types or non-transformer architectures?
- While SliM-LLM claims to reduce overall memory usage, the paper lacks clarity on whether SBA and SQC introduce any hidden memory or storage overheads due to structured bit-width allocation. Further detail here would help assess the method’s efficiency.

---

> ### Author Response · Authors · 2024-11-25
>
> > W1: The paper lacks a transparent discussion of potential limitations, such as possible trade-offs between compression rate and inference speed, or cases where the approach may not perform as well, such as for models with different architectural peculiarities. Acknowledging these limitations would strengthen the overall rigor of the paper.
>
> Dear Reviewer mxH3,
>
> Thank you very much for your suggestion.
>
> We will include a discussion of the limitations in the revised version. We claim that our main contribution lies in the advantages of memory compression for LLMs and the hardware-friendly nature of data storage. As detailed in Table 5 of our manuscript, we analyzed the actual deployment speed under different memory compression levels. Our mixed-precision method is primarily designed based on recent literature and tailored for most commonly used open-source LLM architectures. However, it is true that it has not been specifically adapted for a few unique LLM models. We will carefully analyze the trade-offs and limitations of our work, particularly regarding compatibility with diverse architectures, and include a reasonable explanation of these limitations in the updated version of the paper.
>
> > W2: The experimental evaluation in this paper primarily focuses on the LLaMA series, with additional results on the OPT model series presented in the appendix. However, there is little discussion or empirical validation on architectures beyond these two families, which share many similarities in their transformer structures. This limited scope raises concerns about the generalizability of SliM-LLM across more diverse architectures. The absence of empirical results on these varied architectures leaves open the question of how well the proposed salience-based quantization strategies would perform in models with different salience and outlier channel dynamics. Extending experiments to a broader set of architectures would strengthen the claim of SliM-LLM’s robustness and applicability across LLMs.
>
> We observed that most LLM architectures share similar foundations, being primarily based on Transformer. For our experiments, we selected models including OPT1.3B, 2.7B, 6.7B, 13B, 30B, and LLaMA1~3 in 7B or 13B, designing various experimental setups for configurations. These selections represent a significant portion of LLMs and align with the experimental setups commonly used in recent quantization and compression research.
>
> However, we find your suggestion highly constructive, as it could help enhance the expressiveness and generalizability of SliM-LLM. In response, we have added experiments involving Gemma2 and Mixtral to Table 1 below in these comments.

---

> ### Author Response · Authors · 2024-11-25
>
> > W3: The proposed SliM-LLM approach is designed based on empirical observations of outlier channels and salience distributions in models like LLaMA. However, recent architectures such as Gemma2 incorporate normalization layers following both attention and feed-forward (FFN) modules, which could alter the distribution and behavior of outlier channels. These normalization layers might dampen or redistribute the significance of certain activations, potentially influencing the effectiveness of salience-based quantization strategies. This variation raises questions about how well SliM-LLM’s bit allocation and salience-weighted calibration would generalize to such architectures, where outlier handling may not align with observations made in models without inter-layer normalization. A more in-depth analysis would help determine if additional adaptations are required to support architectures that use different normalization schemes, as seen in models like Gemma2.
>
> We observed that various LLM architectures [1][2], including the Gemma2 model you mentioned, exhibit similar behavior in activations, specifically the presence of certain outlier channels. Even though different LLMs may use different layer normalization methods or apply normalization at varying positions, the intermediate activations in many LLMs consistently show outlier characteristics.
> Our SBA method, as described in the paper, is specifically designed to handle this pattern through mixed-precision quantization. Therefore, we can reasonably infer that it should achieve the expected results across most LLM architectures. Further experiments on Gemma2, summarized in Table 1 below, indicate that the outlier distribution in Gemma2 aligns with that of most LLMs, with certain channels heavily populated by outliers, and Slim-LLM can still effectively quantize these outliers to ultra-low bit-width values under the Gemma2 architecture.
>
> Table 1 PPL Comparison on Gemma2 and Mixtral
>
> | **Model/Evaluation** | **Method**     | **PPL (wikitext2)** |
> | -------------------- | -------------- | ------------------- |
> | Gemma2-9B            | GPTQ 2-bit     | 186.77              |
> |                      | AWQ 2-bit      | 217.83              |
> |                      | SliM-LLM 2-bit | 26.30               |
> | Mixtral 8\*7B         | GPTQ 2-bit     | 16.38               |
> |                      | AWQ 2-bit      | 3.2e5               |
> |                      | SliM-LLM 2-bit | 7.44                |
>
> [1] Spqr: A sparse-quantized representation for near-lossless llm weight compression. ICLR 2024.
>
> [2] Mitigating the impact of outlier channels for language model quantization with activation regularization. arXiv preprint
>
> > Q1: (1) The paper briefly mentions spatial clustering of salient weights, but how does the model respond to cases where salience does not cluster as neatly?
>
> > (2) Are there adaptive mechanisms in SBA and SQC to handle such irregularities?
>
> (1) In Section 3.1, we theoretically demonstrate that outlier activation channels in LLMs have a structured impact on the salience distribution of weights. Since outlier activation channels are consistently present in LLMs, this structured distribution phenomenon is generally observed.
>
> (2) SBA addresses the inconsistency in salience distribution across groups by employing a search-based approach to allocate the optimal bit-width configuration, achieving excellent results under ultra-low bit-width settings.
>
> Additionally, as described in Section 3.3.2, we also observed that within a group, there are still a few discretely distributed salient weights. To address this, we introduced SQC in Equation (5) of our manuscript to reduce quantization errors caused by these discrete salient weights during the quantization process. This effectively resolves the issue of unevenly distributed salience. This observation forms the core motivation for the SBA and SQC methods proposed in our work.

---

> ### Author Response · Authors · 2024-11-25
>
> > Q2: (1)Given the additional steps introduced in SBA and SQC, how does the computational cost compare to baseline quantization methods?
>
> > (2) Specifically, are there latency trade-offs that should be considered, especially for real-time applications?
>
> (1) In our mixed-precision quantization strategy, SBA and SQC are offline quantization steps that convert the original FP16 weights into ultra-low-bit integer values for storage. Both methods are executed on a resource-limited single GPU, and the quantization process for a 7B model takes approximately 50 minutes, which is about the same level as baseline. This is significantly more efficient compared to QAT-based methods, which often require multiple GPUs and can take a day or longer to complete.
>
> (2) While the cost of quantization is worth considering, it is typically not the most critical factor since it is performed offline and can be scheduled with ample time in advance. In real deployment scenarios, inference efficiency is the key priority. As shown in Table 5 in Section 4.3, our method achieves faster inference speeds for the 13B model compared to FP16 while also delivering better performance than GPTQ[1]. This improvement is achieved through the development of CUDA code tailored to the computational characteristics of SliM-LLM, minimizing the overhead of extra computations. Additionally, we detail in the appendix the methods of our data packaging and distribution process, including how quantized data is aligned and packed into a unified data type, as well as how it is unpacked and dequantized during computation.
>
> [1]Frantar, Elias, et al. "Gptq: Accurate post-training quantization for generative pre-trained transformers." arXiv preprint arXiv:2210.17323 (2022).
>
> > Q3: The paper focuses on transformers for text, but given the growing interest in multi-modal LLMs, could this quantization approach generalize to models that handle other data types or non-transformer architectures?
>
> Although our current research focuses on Text-Transformers, as noted in [1-3], most mainstream LLMs and multi-modal LLMs are designed based on Transformer architectures. Their weights are still primarily concentrated in the linear layers of the multi-head attention and feedforward networks (FFN). Moreover, we have observed that while multi-modal LLMs introduce a visual modality to activations, this does not affect the occurrence of outlier channels during inference. Our proposed SliM-LLM is specifically designed to address the structured salience distribution caused by outlier activations. We believe that SliM-LLM holds significant potential for adaptation to multi-modal architectures, and exploring this application will be a key focus of our future research.
>
> [1] Liu, Haotian, et al. "Visual instruction tuning." Advances in neural information processing systems 36 (2024).
>
> [2] Alayrac, Jean-Baptiste, et al. "Flamingo: a visual language model for few-shot learning." Advances in neural information processing systems 35 (2022): 23716-23736.
>
> [3]Li, Junnan, et al. "Blip-2: Bootstrapping language-image pre-training with frozen image encoders and large language models." International conference on machine learning. PMLR, 2023.
>
> > Q4:  While SliM-LLM claims to reduce overall memory usage, the paper lacks clarity on whether SBA and SQC introduce any hidden memory or storage overheads due to structured bit-width allocation. Further detail here would help assess the method’s efficiency.
>
> We emphasize that the overhead introduced by SBA for structured mixed-precision quantization during storage and inference is negligible. This is a key advantage of structured mixed-precision quantization compared to traditional unstructured approaches. As highlighted in Section 3, while we do introduce a small number of structured parameters to indicate the quantization bit-width of each group, the additional parameter cost is minimal due to the design of our structured quantization method.
>
> For example, for LLaMA-7B with a weight matrix dimension of $4096\times 4096$ and a group size of 128, each group has a size of $4096\times 128$. We only require a 4-bit identifier to store the quantization precision of each group, resulting in an overhead of just 0.000008 bits. Compared to other mixed-precision methods, the parameter increase introduced by our structured mixed-precision quantization is entirely negligible.

---

> > ### Comment · Reviewer_mxH3 · 2024-11-26
> > **Response to the authors feedback.**
> >
> > Thank you for the thoughtful revisions and comprehensive responses. I appreciate the effort to address the points raised in the initial review. I have decided to raise my score, believing that the authors will make the necessary adjustments to further enhance the clarity and comprehensiveness of the paper.

---

> > > ### Author Response · Authors · 2024-11-26
> > > **Thanks to Reviewer**
> > >
> > > Dear Reviewer mxH3,
> > >
> > > Thank you for your positive response and for improving the score! We will make the modifications and additions in the final version.

---

### Meta-Review · Area_Chair_933U · 2024-12-20

**Metareview:**

This paper introduces SliM-LLM, a novel salience-driven mixed-precision quantization technique for large language models (LLMs). The method aims to optimize memory efficiency while maintaining model performance by adaptively assigning bit-widths to weight groups based on their salience and fine-tuning quantizer parameters. Extensive experiments on LLaMA models show that the approach achieves significant memory reduction and improved accuracy, particularly for ultra-low bit-width quantization, outperforming state-of-the-art techniques in terms of both efficiency and performance.

One of the key strengths of this paper lies in its innovative approach to mixed-precision quantization, which combines salience-driven bit allocation and calibration. The method demonstrates strong results, particularly in terms of memory efficiency and model performance, with clear experimental evidence supporting its effectiveness. The paper is well-written, and the visualizations provided help to clarify the method and results, making the contribution accessible to the reader.

However, several concerns raised by the reviewers remain unresolved. The primary issue centers around the performance trade-offs associated with the proposed 2-bit quantization method. Despite achieving a 16x reduction in memory, the method leads to a noticeable increase in perplexity and slower throughput compared to FP16. While the authors provided clarifications in response, the reviewers were not fully convinced that the proposed approach consistently outperforms alternatives, especially in terms of throughput and perplexity. Additionally, the experimental validation focuses mainly on the LLaMA and OPT model families, which are similar in architecture. The lack of evaluation on a broader set of models raises concerns about the method's generalizability.

In conclusion, while the paper presents an interesting and innovative approach to quantization, the unresolved issues regarding performance trade-offs and limited generalization to other architectures lead to a recommendation for rejection. The memory efficiency improvements are promising, but the increase in perplexity and slower throughput diminish the method's practical applicability. Furthermore, the insufficient exploration of the approach’s generalization beyond LLaMA and OPT models, along with a lack of detailed theoretical analysis, limits the overall contribution of the paper.

**Additional Comments On Reviewer Discussion:**

During the rebuttal period, the reviewers raised concerns primarily around two issues: performance trade-offs and the generalization of the proposed method. One reviewer highlighted the increase in perplexity and slower throughput when using the proposed 2-bit quantization compared to FP16, despite the substantial memory reduction. They questioned whether the method could consistently outperform alternatives, especially in terms of efficiency and throughput. The authors addressed this by clarifying the experimental setup and providing additional data. However, the reviewer remained unconvinced, particularly due to the performance degradation observed with the 2-bit method.

The second major concern was the limited generalization of the approach, as experiments were primarily conducted on LLaMA and OPT models, which share similar architectures. The authors expanded the experimental results to include additional models, but the reviewer still expressed reservations about the method's applicability to a wider range of architectures. Although the authors acknowledged this limitation, they did not fully alleviate the concerns about the method's broader applicability.

In my final decision, I took these unresolved issues into account. While the authors made efforts to address the points raised, the lack of convincing evidence on performance consistency and the narrow experimental scope led me to believe that the method’s practical utility is still uncertain. Therefore, I maintained my recommendation for rejection.

---

### Decision · Program_Chairs · 2025-01-22

Reject